# Preparation and Application Progress of Imprinted Polymers

**DOI:** 10.3390/polym15102344

**Published:** 2023-05-17

**Authors:** Yongsheng Shen, Pengpai Miao, Shucheng Liu, Jie Gao, Xiaobing Han, Yuan Zhao, Tao Chen

**Affiliations:** 1Hubei Key Laboratory of Radiation Chemistry and Functional Materials, School of Pharmacy, School of Nuclear Technology and Chemistry & Biology, Hubei University of Science and Technology, Xianning 437100, China; m13177439906@163.com (Y.S.); zzu666666@126.com (P.M.); hanxiaobing@hbust.edu.cn (X.H.); zhyf308@hbust.edu.cn (Y.Z.); 2Institute of Forensic Science, Hunan Provincial Public Security Bureau, Changsha 410001, China; 13875861375@163.com

**Keywords:** imprinted polymers, principle, classification, preparation, application

## Abstract

Due to the specific recognition performance, imprinted polymers have been widely investigated and applied in the field of separation and detection. Based on the introduction of the imprinting principles, the classification of imprinted polymers (bulk imprinting, surface imprinting, and epitope imprinting) are summarized according to their structure first. Secondly, the preparation methods of imprinted polymers are summarized in detail, including traditional thermal polymerization, novel radiation polymerization, and green polymerization. Then, the practical applications of imprinted polymers for the selective recognition of different substrates, such as metal ions, organic molecules, and biological macromolecules, are systematically summarized. Finally, the existing problems in its preparation and application are summarized, and its prospects have been prospected.

## 1. Introduction

Sensitive sensing of the surrounding environment has become necessary for modern life. Molecular recognition is the foundation of sense, which is very important to some biological processes and is the focus of much material investigation due to its importance in sensing processes, separations, and detection [1]. Based on the natural antibody–antigen, enzyme–substrate recognition systems and synthetic receptors with selectivity were developed, namely imprinted polymers [2]. Imprinted polymers are porous materials that are prepared via imprinting technology, which can completely match templates in cavity structure and possess specific recognition functional groups/binding sites for template molecules [3]. In 1972, the Wulff group first proposed the concept of molecularly imprinted polymers (MIPs) [4]. In 1993, the theophylline-based MIPs were reported by the group of Vlatakis in the journal of *Nature* [5]. Since then, MIPs have achieved rapid development, and great progress has been made in the types, synthesis methods, and applications of MIPs, as well as the principles of molecularly imprinted technology [6].

Recently, MIPs have been extensively used in many fields such as sample preparation, analysis and detection, environment protection, and drug release [7,8,9]. With the application expansion of MIPs, the requirements for structural design and preparation methods are increasing, leading to more and more MIPs with different structures being prepared by different methods [10,11,12]. Though many reviews of MIPs about specific aspects have been reported, there are few reviews about the classification, preparation, and application [13,14,15,16,17,18,19]. With the rapid development and application of new MIPs, now is an appropriate time to summarize the recent progress [20,21,22,23,24,25,26].

## 2. Recognition Principles of Imprinted Polymers

MIPs are versatile functional materials that not only have adsorption properties but also have excellent recognition [11,12]. The specific recognition of MIPs is closely related to their cavity structure and intermolecular interactions (Figure 1) [27], including (A) reversible covalent bonds, (B) semi-covalent bonds, (C) electrostatic attraction, (D) van der Waals interactions, and (E) metal coordination.

In the process of MIPs preparation [28], template-functional monomer complexes are formed firstly through intermolecular interactions, achieving a high degree of binding site matching. Then, a cross-linking agent is added to fix the functional group in space via polymerization, as the polymers are grown around template molecules; thus, precise cavity matching can be achieved in this process. Finally, a three-dimensional cavity with specific recognition for the template can be obtained by removing the template.

## 3. Classification of Imprinted Polymers

With the deepening of research, the types of MIPs are increasing. According to the binding mode between template molecules and functional monomers, the MIPs are divided into covalent/pre-assembled MIPs and non-covalent/assembled MIPs [12]. As the difference of substrates, the MIPs can also be divided into polymers that recognize ions, organic molecules, and biological macromolecules [11]. In addition, according to the distribution position of templates in imprinted polymers, they can also be divided into bulk imprinted polymers, surface imprinted polymers, and epitope imprinted polymers [29].

The distribution of templates in imprinted polymers has a significant influence on the removal and recognition rate of templates, which determines the application of the imprinted polymers (Table 1) [30,31]. For the bulk imprinted polymers, templates are randomly distributed in the bulk of the matrix. The template near the center has a slow removal and recognition rate, so they can only be used for the selective recognition of metal ions or organic molecules. For surface imprinted polymers [32,33], templates are located in the surface layer, making the fast removal and recognition of templates, which can be used for the recognition of most substrates. For the epitope imprinted polymers, only the exposed chain segments in the macromolecules are used as templates in the MIPs preparation, resulting in a very fast removal and recognition of templates, making them very suitable for the recognition of biological macromolecules [31].

### 3.1. Bulk Imprinted Polymers (BIPs)

BIPs are the earliest products obtained with molecular imprinting technology. Typical preparation methods of BIPs are as follows: the templates and functional monomers are mixed to form complexes; then, cross-linkers and initiators are added to the mixture; after the polymerization of the system, the BIPs are obtained with mechanically ground and sieved [6].

A rubidium (Rb) ion imprinted polymer was synthesized via bulk polymerization using Rb^+^ ion as template [34], crown ether as ligand, methacrylic acid (MAA) as functional monomer, ethylene glycol dimethacrylate (EGDMA) as cross-linker, and azobisisobutyronitrile (AIBN) as initiator. The maximum adsorption capacity of these BIPs was 213 mg/g, and they exhibit excellent selectivity for Rb^+^ ions compared with Li^+^, Na^+^, and K^+^ ions. A novel biosensor combining MIPs and Raman spectroscopy was developed to determine melamine in milk [35]. MIPs were prepared via bulk polymerization of the template, MAA, EGDMA, and AIBN. The detection limit of the obtained MIPs is 0.012 mmol/L, and the detection time is reduced to less than 20 min in whole milk.

The reaction process of BIPs is the simplest, but the post-processing is complicated and needs crushing and grinding. In addition, the complete removal of templates is very difficult in BIPs, especially for templates distributed near the center. This can not only slow the elution rate but also affects the adsorption efficiency. Therefore, imprinting technology that establishes active sites on the surface layer of imprinted materials has been extensively investigated.

### 3.2. Surface Imprinted Polymers (SIPs)

SIPs are prepared with two main steps: modification of carrier and polymerization [40]. Typically, spherical or layered carriers such as silica, metal oxides, and nanomaterials are modified with coupling agents, making it complex with template and polymerization with cross-linker. Then, mixed with templates, functional monomers, and initiators, after polymerization, the SIPs can be obtained. The recognition site of SIPs is located in the surface layer, which can solve the problems caused by BIPs, such as deep embedding and the difficult removal of templates [41].

In the work of Song and co-workers [36], bovine serum albumin (BSA) was first immobilized on the anchored tetraalkylammonium groups of the poly (VBDC-CMS) nanoparticles via anion exchange interactions, and the thickness-controlled imprinted film was obtained with surface-initiated polymerization. Core–shell structural magnetic molecular imprinted polymer (MMIP) with surface imprinted technology was reported for the recognition of gallic acid (Figure 2) [37], and hollow magnetic molecular imprinted polymer (HMMIP) was synthesized by etching the intermediate silica layer of MMIP. The results showed HMMIP had higher selectivity and adsorption capacity towards gallic acid than MMIP.

### 3.3. Epitope-Imprinted Polymers (EIPs)

EIPs are also known as antigenic determinant imprinted polymers; the antigenic determinants refer to the segments of antigen macromolecules that can be recognized by antibodies. Inspired by the specific recognition between antibodies and antigens, novel EIPs were developed [42]. The preparation of EIPs is similar to that of SIPs, while only the exposed chain segments rather than the biological macromolecules were used as templates. The resulting imprinted materials can not only recognize the chain segments but also can bind the entire biomacromolecules [43,44]. Using the chain segments as templates for the MIPs preparation can reduce the impact of non-specific adsorption components in biomacromolecules. In addition, the chain segments are relatively stable in reaction conditions, which can avoid the destructive effect of harsh reagents on biomacromolecules with traditional methods [45,46]. For the simple and early detection of COVID-19, EIPs based on SARS-CoV-2 spike protein subunit S1 (ncovS1) were fabricated and applied in an electrochemical sensor [38]. The obtained sensor showed a short reaction time of 15 min and can detect ncovS1 both in a buffer solution and in the patient’s nasopharyngeal. A helical peptide was selected from the HIV protease as a template (Figure 3) [39], mixed with acrylic acid, N-benzyl acrylamide, acrylamide, and N,N-ethylene-bis-acrylamide; imprinted polymers were obtained with UV irradiated polymerization, after the removal of peptide, the EIPs shows highly recognition for HIV.

## 4. Preparation Progress of Imprinted Polymers

MIPs are mainly prepared with free radical polymerization of template-functional monomers and cross-linking agents [7,8,9], which can be realized with bulk polymerization, emulsion polymerization, solution polymerization, suspension polymerization, etc. The MIPs are always prepared with thermal polymerization and in the presence of an initiator, which needs a long reaction time and high energy consumption. To solve the problems that appeared in traditional thermal polymerization, novel radiation polymerization technologies such as UV radiation, γ-rays radiation, electron beam, and microwave have been used in the synthesis of MIPs [47]. Reaction time can be shortened via these radiation polymerizations, and initiators were not required for some radiation polymerization. In addition, plenty of organic solvents were used in the traditional approach, which will cause environmental pollution and safety risk. To solve environmental and safety issues, besides the traditional bulk polymerization, green strategies based on supercritical carbon dioxide, ionic liquids, and deep eutectic solvents have also achieved rapid development [3].

### 4.1. Thermal Polymerization

The main and conventional approach for the preparation of MIPs is thermal polymerization, including conventional thermal polymerization and thermal polymerization with new technology (Table 2) [48]. Though long reaction time and the high energy consumption is needed for thermal polymerization, they can be used for the preparation of most MIPs, including bulk, surface, and epitope imprinted polymer.

#### 4.1.1. Conventional Thermal Polymerization

MIPs for benzylpiperazine were developed through thermal polymerization via both self-assembly and semi-covalent methods, and the recognition abilities of the obtained MIPs were compared [49]. For the self-assembly approach, benzylpiperazine and methyl methacrylate form complexes through self-assembly. While for the semi-covalent approach, the benzylpiperazine reacts with vinylbenzenesulfonyl chloride to form new molecules. Magnetic SIPs using tolfenpyrad as a template were synthesized through thermal polymerization with 2-vinylpyridine and ethylene magnetite nanoparticles (Figure 4) [50]. The maximum adsorption capacity of the MIPs toward the target was 7.20 mg/g and exhibited excellent selectivity.

#### 4.1.2. Other Thermal Polymerization

Thermal polymerization is always carried out through liquid media for heat transfer; now, heat transfer with other approaches for thermal polymerization was reported for the MIPs preparation. Quercetin-based SIPs were successfully prepared via thermal polymerization with an oven using AA and EGDMA as functional monomers and cross-linking agents, respectively [51]. A rapid and direct strategy for thermal polymerization of polymer shells at magnetic particle surface with an alternating magnetic field was developed for the preparation of MIPs of *para*-nitrophenol at room temperature [52]. This method provides a strategy for magnetic particle surface imprinting without obvious temperature increases.

### 4.2. Radiation Polymerization

Many issues appeared in the traditional thermal polymerization for the preparation of MIPs. On the one hand, a long time was needed, and the efficiency was always low. On the other hand, the high reaction temperature needs more energy and also affects the stability of the complexes formed by template and functional monomer [7,8,9]. In addition, the use of additives such as initiators, dispersants, and emulsifiers causes the issue of residues, leading to the obtained MIPs not being suitable for application in the medical area. In order to solve these issues, novel radiation polymerization technologies have been rapidly developed (Table 3) [47]. Radiation polymerization can not only shorten the reaction time but also can be conducted at room temperature. Moreover, some radiation polymerization does not even require an initiator, which can be used for the preparation of MIPs used in the medical area [53].

#### 4.2.1. UV Radiation Polymerization

Due to the easy accessibility of radiation equipment, UV radiation became the most popular radiation approach for the synthesis of MIPs. Compared with traditional thermal polymerization, UV radiation polymerization does not need a high reaction temperature. For the low energy of UV radiation, it is always necessary to add an initiator. Novel benzyl mercaptan BIPs with controlled binding sites were synthesized via thermal polymerization (80 °C) and UV radiation polymerization at room temperature [54]; the template-binding strengths of the two methods are 39% and 67.5%, respectively, demonstrating the better recognition performance of MIPs obtained with UV radiation polymerization. MIPs for glutathione were prepared through controlled radical polymerization under UV radiation at room temperature [55]; spherical shape, fast binding kinetic, and highly selective factor was obtained for UV polymerization than those of traditional thermal polymerization, and the proposed method was successfully used to determined glutathione in spiked human urine.

The penicillin residues in the environment cause health risks and increase the development of resistances, so its selective recognition from complex matrices is challenging work. MIPs for the recognition of penicillin G were prepared through UV polymerization as follows [56]: template was added to the acetonitrile, followed by the addition of MAA and trimethylolpropane trimethacrylate, nitrogen was purged for 15 min, and AIBN was added; the photopolymerization was conducted with high-pressure mercury. UV radiation polymerization has the merit of being conducted at a lower temperature, which can prevent the degradation of the substrate. UV self-initiated MIPs based on Au and TiO_2_ functionalized metal-organic framework has been first developed for the selective separation of caffeic acid [57]; the preparation needs no initiator. The polymerization could be initiated by hydroxyl radicals, which are generated by TiO_2_ under UV light. This is a more eco-friendly and more efficient method than traditional thermal polymerization. Atrazine MIPs were synthesized by far-infrared and UV radiation polymerization (Figure 5) [58]. Compared with commercial solid phase extraction sorbent, higher recoveries of atrazine in practical water samples were obtained for both MIPs. In some cases, UV radiation polymerization can be carried out without the addition of an initiator, such as photocatalytic performance observed for the carriers.

#### 4.2.2. γ-rays Radiation Polymerization

γ-rays are generally produced by the decay of radioactive elements such as ^60^Co, which has a short wavelength and high frequency, resulting in higher energy and penetration. No significant changes in temperature can be observed in γ radiation polymerization, and no initiator is required [47]. Reversible addition-fragmentation chain transfer (RAFT) polymerization was utilized to imprint atrazine [59] onto porous fabric via grafting polymerization of MAA, which uses γ-rays for the generation of radicals. The positron annihilation lifetime spectroscopy (PALS) results showed that monomer/chain transfer agent concentration ratios are effective for the formation of template cavities. In addition, well-defined erythromycin imprinted porous polythylene/polypropylene nonwoven fabrics were also prepared with γ-rays radiation-induced RAFT-mediated polymerization [60], and the greatest binding energy appeared at an MMA/erythromycin ratio of 4:1.

Zsebi and co-workers reported that MIPs use phenytion as a template, which has been synthesized by γ-rays radiation copolymerization of acrylamide and EGDMA [61]. With the increase in radiation dose, the imprinting factor of the obtained MIPs first increases and then decreases, and the maximum imprinting factor is 2.5 with a radiation dose of 15 kGy. MIPs were prepared by γ-rays radiation polymerization, with bacitracin used as the target molecule [62]. The MIPs materials showed higher adsorption capacity than non-printed polymers and showed a greater selective recognition of bacitracin than other substrates. γ-rays radiation-induced MIPs of glucose were reported by Djourelov’s group, and the effect of cross-linker types and template amounts on the imprint quantities was investigated [63]. The PALS results revealed that cavity size could be controlled with cross-linker concentration and size, template/monomer ratio, and dose of irradiation. For the preparation of errium ion imprinted polymers, a prepolymer complex was synthesized first and then copolymerized via γ-rays radiation polymerization with different kinds of functional monomer and cross-linking agents [64]. The obtained imprinted polymer has an enrichment factor of 25 and could separate erbium from competitive ions such as yttrium, dysprosium, holmium, and thulium. Due to the difficulty in monitoring trace quantities of steroids in a biological environment, MIPs were obtained with thermal, UV, and γ-rays radiation polymerization and used for preconcentration and cleanup of these hormones [65].

#### 4.2.3. Electron Beam Radiation Polymerization

Electron beam (EB) radiation polymerization is conducted under high-energy electron beams, which originated from electron accelerators. The high-energy electrons can interact with substances to generate free radicals, thereby triggering chain polymerization. The energy loss of EB radiation polymerization is only 1/40 of thermal polymerization and will not cause environmental pollution [66]. Baicalin MIPs was prepared with EB radiation polymerization for the first time by the Liu group; no initiator was needed for this EB radiation polymerization [66]. This approach has a very high efficiency, which opens up a new path for the industrial application of MIPs. A porous imprinted membrane was synthesized with EB radiation polymerization; the obtained MIPs are suitable for the recognition of ibuprofen [67]. Ibuprofen was first coordinated with metal ions as a template, and the MAA and EGDMA were added; after EB radiation polymerization and removal of the template, the obtained MIPs have excellent selectivity for the chiral ibuprofen. Chloramphenicol imprinted microspheres of uniform size were obtained with rapid precipitation polymerization, which was initiated via EB radiation [68]. The results show that EB radiation can successfully imprint chloramphenicol in a three-dimensional network structure, and MIPs with a high specific surface area can be obtained. Wang et al. Successfully prepared quercetin-nickel coordination MIPs by EB radiation polymerization [69], the obtained MIPs showed significant selectivity for the complex, with the highest adsorption capacity of 82. 22 μmol/g.

In the work of Liu and co-workers, BIPs were prepared by EB radiation polymerization using sulfamethazine as a template, acrylamide as a functional monomer, and EGDMA as a cross-linker [70,71]. The results showed that excellent recognition performance of the MIPs was obtained at 150 kGy, and the corresponding imprinting factor was as high as 12.91. Thorium ion imprinted polymers were reported by Selambakkannu et al. using EB radiation grafting polymerization [72]; the resulting functional nonwoven fabrics not only have high adsorption capacity but also have good selectivity towards thorium ions.

#### 4.2.4. Other Radiation Polymerization

Besides UV, γ-rays, and EB radiation polymerization, new radiation polymerization based on microwave, ultrasound, and plasma has been developed in the preparation of MIPs in recent years.

A simple and fast polymerization strategy was developed for the synthesis of olivetol MIPs [73]. With the help of microwave radiation, polymerization was carried out in 5.5 min, and the recovery of olivetol, *m*-toluidine, and phenol for the MIPs were 87.3–93.6%, 18.9–24.9%, and 21.4–27.2%. Dimethyl phthalate (DMP) MIPs were prepared with the same approach; the selectivity coefficients were 5.6, 2.6, and 1.4 for DMP, dibutyl phthalate, and dioctyl phthalate [74]. The microwave-assisted approach was also developed for the rapid preparation of bisphenol A MIPs; compared with thermal polymerization, the proposed approach shortened the reaction time 20-fold [75]. The obtained MIPs fiber was used for selective extraction of bisphenol A, diethylstilbestrol, and hexestrol in tap water. Gibberellin acid (GA) magnetic MIPs beads were fabricated via microwave radiation polymerization [76], and the obtained beads were used for the selective separation of GA in real samples. The GA MIPs beads exhibited selective absorption behavior for the target template and showed higher adsorption capacity (708.4 pmol). Quercetin SIPs were prepared using silica as a carrier under microwave radiation [77]; the obtained results showed that this method could greatly shorten the reaction time, and the maximum adsorption for the template is 2.87 μmol/g. Chromium ion imprinted polypropylene fibers were synthesized via the plasma-mediated grafting method (Figure 6) [78], and the obtained imprinted materials exhibited excellent selectivity to chromium ion compared with non-imprinted fibers.

Naphthol SIPs were fabricated by ultrasonic radiation polymerization with attapulgite as a matrix, and acrylic acid modified β-cyclodextrin as a functional monomer [79]. Compared to polymers synthesized by thermal polymerization, the MIPs prepared by ultrasonic radiation had higher selectivity and faster adsorption rate to different templates. MIPs based on ultrasound-assisted polymerization were used for the separation of caffeine [80], and microspheres with uniform size distributions were obtained with the specificity of the template. In addition, magnetic caffeine MIPs microspheres were also reported by Phutthawong’s group [81], with the same ultrasound-assisted precipitation polymerization.

### 4.3. Green Polymerization

Organic solvents are always used in the synthesis of MIPs with conventional approaches, which not only cause environmental pollution but also pose potential safety hazards [10,11,12]. To solve these problems, many green approaches were developed for the synthesis of MIPs, including supercritical CO_2_, ionic liquids, and deep eutectic solvents, as well as bulk polymerization and sol–gel polymerization (Table 4) [3].

#### 4.3.1. Bulk Polymerization

Bulk polymerization is the simplest and most environmentally friendly approach for the preparation of MIPs [102]; only template molecules, functional monomer, cross-linking agent, and initiator (not necessary for γ-rays and EB radiation polymerization) were needed in this polymerization, which has been widely used in the preparation of powder BIPs. Sulpiride MIPs were prepared by bulk polymerization with itaconic acid (ITA) as a functional monomer [82]. The obtained MIPs were demonstrated with the molar ratio of sulpiride/ITA/EGDMA of 1/4/15, and the obtained MIPs showed good performance with a high imprinting factor of 5.36 and a maximum adsorption capacity of 61.13 μmol/g. MIPs based on methacrylic acid functionalized β-cyclodextrin monomer was prepared through bulk polymerization for the selective recognition of benzyl paraben (Figure 7) [83]. The functional monomer shows strong interactions with the template, including π-π stacking, inclusion complex, and hydrogen bonding. In addition, the binding experiment results revealed that the β-cyclodextrin could significantly enhance the recognition affinity.

Basic Blue dye MIPs were synthesized with bulk polymerization and used for the adsorption of Basic Blue in wastewater [84]. The Basic Blue adsorption on the MIPs obeyed the second-order kinetic model and the Langmuir isotherms model, with maximum adsorption capacities of 99.0 mg/g. Bisphenol A MIPs was synthesized via bulk polymerization using phenolphthalein as a dummy template [85]. MIPs particles with a diameter of 40–60 μm and a high surface area of 359.8 m^2^/g were obtained; the obtained MIPs showed specific adsorption for bisphenol A. Frontal polymerization was successfully applied for the first time in the fabrication of levofloxacin MIPs [86], this approach offered a solventless polymerization, and the reaction time can be shortened to 30 min.

Though bulk polymerization can be widely used in the green synthesis of powder MIPs, it is powerless for the preparation of spherical MIPs or SIPs, which need dispersion media. Therefore, it is necessary to develop green reaction media for the synthesis of MIPs.

#### 4.3.2. Supercritical Carbon Dioxide Polymerization

As it has many advantages, such as being non-flammable, non-toxic, and easily removed, supercritical carbon dioxide (scCO_2_) became an environmentally friendly alternative to traditional organic solvents [103]. Viveiros et al. developed a cheap acrylate and acrylamide-based copolymer in scCO_2_, methacrylamide-based MIP showed the maximum adsorption capacity for model pharmaceutical impurity, acetamide [87]. In the further investigation of Viveiros et al., a computational approach was used to optimize the MIP synthesis in scCO_2_. The results showed that itaconic acid and 2-hydroxyethyl methacrylate have strong interactions with acetamide [88].

Gallic acid MIPs were synthesized using scCO_2_ as a green process and MAA and MMA as functional monomers [89]. The template removal rate with Soxhlet extraction was about 95–99%, and the results indicate that the synthesized MIPs possess high selectivity and separation abilities. To reveal the regional distribution of labdanolic acid (LA) [90], a series of MIPs were prepared using scCO_2_ and used for the separation of LA. In all of the obtained MIPs, 2-(dimethylamino)ethyl methacrylate MIPs exhibited the optimal result for LA purification. A novel class of flufenamic acid (FA) MIPs was also synthesized using the scCO_2_ technique [91], the NMR experiments confirmed that the main interaction that exists between the FA and functional monomer is hydrogen bonds.

#### 4.3.3. Ionic Liquids Polymerization

Ionic liquids (ILs) are organic salts consisting of anions and cations, which can remain liquid at ambient temperature. Compared with traditional organic solvents, ILs exhibit many advantages such as being non-flammable, having excellent thermal stability, and being non-volatile, completely recycled, leading to their wide application in the synthesis of polymers [104]. It has been shown that synthesizing MIPs in ILs can accelerate the reaction, enhancing the recognition and adsorption capacity. Nowadays, besides solvents, ILs can also be used as templates, monomers, cross-linkers, and additives for polymerization [3].

Isoquercitrin (ISO) MIPs were synthesized in ILs, using 4-vinylpyridine as a functional monomer [92]. The optimal recovery of obtained MIPs for ISO was 87.78%. Naproxon MIPs were synthesized with a mixture of polyhedral oligomeric silsesquioxane (POSS) substituted MA, 4-vinylpyridine, and EGDMA in ILs of [BMIM]BF_4_ [93]. The highest imprinting factor of the MIPs synthesized with modified MA was 22, which is much higher than that synthesized without POSS. Methyl gallate (MG) MIPs was also prepared in the same ILs; the highest imprinting factor is 10.9 at the optimized polymerization parameters [94]. Propranolol MIPs were synthesized by Booker’s group with four different ILs, and the cavity size of the obtained MIPs was revealed by PALS [95]. Aconitic acid MIPs were also prepared in ILs by Booker’s group; under UV radiation and thermal conditions giving polymer microspheres. Compared with traditional polymerization in acetonitrile, higher selectivity indices were obtained for the MIPs synthesized in ILs [96].

#### 4.3.4. Deep Eutectic Solvent Polymerization

Deep eutectic solvents (DESs) are considered the fourth generation of ILs, which are salt mixtures obtained with the complexation of hydrogen acceptor and naturally derived uncharged hydrogen bond donors [105]. DESs can also be made from non-ionic compounds, which is better than conventional ILs. In addition, DESs also possess the merits of low cost, low toxicity, biodegradability, and no need for additional purification.

EDS-modified MIPs were synthesized for the recognition of chlorogenic acid (CA) [97]; the extraction rate of CA was 12.57 mg/g. Magnetic DESs MIP was synthesized for the recognition and separation of Bovine hemoglobin [98], and the maximum adsorption capacity was calculated with the Langmuir isotherms model to be 175.44 mg/g. In addition, the imprinted materials showed a high imprinted factor of 4.77, which presented outstanding recognition specificity. Gallic acid MIPs were synthesized with bulk polymerization using DESs as functional monomers (Figure 8) [99], the obtained MIPs has a mesoporous structure with an average pore diameter of 9.65 nm. The adsorption behavior followed pseudo-second-order kinetic model, with a maximum adsorption capacity of 0.711 mmol/g.

#### 4.3.5. Sol-Gel Polymerization

In the separation investigation of gossypol, the sol-gel method was applied for the synthesis of SIPs [29]. Typically, gossypol was dissolved in acetone, and (3-aminopropyl) triethoxysilane was added to the mixture to obtain self-assemble complexes. Then, activated silica carrier, cross-linker tetraethoxysilane, and acetic acid were added. Finally, SIPs were obtained with reaction at room temperature for 24 h. Results revealed that the MIPs obtained with sol-gel polymerization was a desirable sorbent for rapid adsorption of gossypol, and the MIPs obtained with bulk polymerization was suitable for selective recognition of gossypol. A paper-based fluorescent senor targeting glyphosate, integrated with surface imprinting technology, was reported by Wang’s group [100]. SIPs were prepared via sol–gel polymerization, and the obtained SIPs exhibited high selectivity for glyphosate. The detection accuracy of the obtained sensor was relatively good, with a recovery rate of 92–117% for practical samples. The sol–gel approach was used for the fabrication of quercetin-based SIPs (Figure 9) [101], the adsorption could reach equilibrium within 90 min, with a maximum adsorption capacity of 35.7 mg/g. The mechanism for adsorption isotherm and kinetics of SIPs was proved to obey the Freunflich isotherm model and pseudo-second-order kinetics model.

## 5. Application Progress of Imprinted Polymers

Due to their excellent recognition, structure-activity prediction, and widespread applicability, the MIPs have achieved rapid progress [10,11,12]. Based on their strong recognition ability, high selectivity, and good stability, MIPs are extensively used in various fields such as solid phase extraction and analytical detection. In addition, MIPs are expected to achieve industrial production and application in environmental monitoring, the food industry, clinical medicine, natural products, and other industries [106,107,108,109]. According to the difference in templates, the latest progress in the recognition applications of MIPs in metal ions, organic molecules, and biological macromolecules was summarized.

### 5.1. Ion-Imprinted Polymers (IIPs)

IIPs are polymers prepared through imprinting technology using ions as templates, which have been extensively applied for the reparation and determination of heavy metals in wastewater [110]. Due to the small radius of ions, for the preparation of IIPs, polymerizable ligands are always needed for the formation of template-functional monomer complexes (Table 5).

Novel nanostructured magnetic IIPs were prepared for the selective separation of Pb(II), using ITA as a functional monomer [111]. The fast adsorption rate revealed the Langmuir adsorption and second-order kinetic model, and the selective factors of Pb(II)/Co(II), Pb(II)/Cu(II), Pb(II)/Zn(II) were 45.6, 6.45, and 8.3, respectively. Cd(II) ions were first imprinted within modified chitosan, which has been further grafted to magnetic silica [112]. The highest adsorption capacity for Cd(II) was 26.1 mg/g, and the selective factors of Cd(II)/Cu(II), Cd(II)/Cr(II), Cd(II)/Pb(II) were 3.315, 3.875, and 2.061. Magnetic Cr(III) IIPs were prepared using Cr_2_O_7_^2−^ as a template, 4-vinylpyridine as a monomer [113]. The highest adsorption capacity of Cr(III) was 201.55 mg/g, and the adsorption capacity decreased by only 8.2% after being reused six times. As(III) IIPs were prepared via bulk polymerization using 3-mercaptopropyl trimethoxy-silane and dithioerythritol as monomers [114], the obtained IIPs showed a good spherical structure with a high surface area of 779.80 m^2^/g. In the treatment of practical water samples, the IIPs showed a recovery rate of 95.0–105.0%. Novel magnetic Hg(II) IIPs with Fe_3_O_4_@SiO_2_ incorporation were prepared through surface imprinting (Figure 10) [115], using allyltiourea as a coordinated monomer. The optimal adsorption capacity for Hg(II) was as high as 78.3 mg/g, and the relative selectivity factor of Hg(II)/Ni(II), Hg(II)/Cu(II), Hg(II)/Co(II), Hg(II)/Cd(II) was 623, 355, 623, and 155, respectively.

Except for the severely toxic heavy metal ions, IIPs have also been applied in the recognition of other ions. For rapid removal of Ni(II) ions from waste, Ni(II) IIPs were prepared by bulk polymerization, and the effect of synthesis parameters on recognition and adsorption properties was investigated [116]. The selective factors of the obtained IIPs for all samples are greater than one, and the obtained IIPs showed high reusability and stability. Zn(II) IIPs were synthesized by free radical polymerization using morin as a ligand and 4-vinylpyridine as a functional monomer [117]. The detection limit of the obtained IIPa was 2.9 μg/L, with a dynamic linear range of 25–200 μg/L. Three functional monomers, including 4-vinylpridine, 2-(allylthio) nicotinic acid, and 2-Acetamidoacrylic acid, were chosen to prepare Pd(II) IIPs [118]; the results of competitive adsorption experiments showed high selectivity for Pd(II). Novel Br(I) IIPs were synthesized for selective separation of Br(I), using modified chitosan as monomer and glutaraldehyde as cross-linker [119], and the optimal adsorption capacity was 18.89 mg/g. The obtained IIPs showed high selectivity for Br(I), and the adsorption process followed the Freundlich isotherm model and second-order kinetic model. Novel Cu(II) phenanthroline(vinyl benzoate)_2_H_2_O complex was synthesized and used to construct new IIPs (Figure 11) [120], the IIPs showed high adsorption capacity of 287.45 mg/g at 1600 mg/L Cu^2+^ ions.

### 5.2. Organic Molecular Imprinted Polymers (OMIPs)

Due to the wide variety and designable of intermolecular interactions, the most widely used imprinting materials are still the recognition and detection of various organic molecules, including synthetic and natural drugs, dyes, chemical materials, additives, etc. [106,107,108,109]. Many OMIPs have been introduced in the section on MIPs classification and preparation; here, we mainly summarize the latest application (Table 6).

A core–shell magnetic MIPs was prepared by the suspension polymerization/surface imprinting technology, which was used for aniline adsorption from textile wastewater [121]. The obtained MIPs showed a detection limit of 1 ng/mL, with good linearity, recovery, and precision. Melamine MIPs were synthesized and embedded into a thermally conductive layer; the obtained sensor exhibited an excellent recognition for melamine, with a detection limit of 6.02 μM [122]. Methotrexate (MTX) magnetic MIPs were synthesized via the sol-gel method, and the MTX adsorption capacity was 39.56 mg/g, with an imprinting factor of 9.40 [123]. Heterocyclic aromatic amine (haa) MIPs nanospheres were prepared via RAFT polymerization (Figure 12) [124]; then, core–shell structural haa-MIPs with hydrophilic shells (MIP-HSs) were synthesized via grafting polymerization. The hydrophobic haa-MIPs cannot recognize harmine in an aqueous solution, with the improvement of hydrophilic, the MIP-HSs showed efficient recognition of harmine in an aqueous solution.

Novel graphene oxide (GO) based MIPs were fabricated and used for selectivity concentration of bis(2-ethylhexyl) phthalate (DEHP), which is a widely used plasticizer in the plastic industry [125]. DEHP was selectivity separated in real water by GO-MIPs under optimized conditions; the enrichment factors are over 100-fold. EIPs were synthesized using the DYKD peptide as a template [126], and the obtained EIPs were used as selective adsorption materials with good recoveries and high selectivity for DYKD and DYKDDDDK peptides. Novel biomimetic magnetic SIPs were reported by Goyal and co-workers, which can be used for the enantioseparation of a chiral drug such as S-naproxen [127]. The highest binding capacity was found to be 127 mg/g, with a high imprinting factor of 12.88. Magnetic natural salidroside MIPs were synthesized via surface imprinting technology (Figure 13) [128], and the efficiency of controlled release and specificity of recognition was investigated. The total amount of salidroside release at 37 °C is 86%, and the release procedure followed Fickian kinetics.

### 5.3. Biomacromolecules Imprinted Polymers (BMIPs)

With the development of surface imprinting technology, especially the emergence of epitope imprinting technology, imprinted polymers have been widely used in the recognition and separation of biological macromolecules (Table 7) [10,11,12]. Instead of imprinting the whole biological macromolecules, imprinting exposed peptides is gaining popularity for its low cost and high stability [129,130,131,132,133].

EIPs were synthesized for three epitope peptides from the epidermal growth factor, realizing the selection of epitopes for diagnostic applications, which has been verified with many proteins [134]. Mesoporous EIPs were prepared to enhance the selective separation of Cytochrome c (Cyt c) with amphiphilic ILs as a surfactant; the exposed nonapeptide of Cyt c was used as the template [135]. The obtained EIPs have an appropriate cavity size, which can promote the mass transfer of Cyt c, leading to a high capacity of 86.47 mg/g. To improve the mechanical property and imprinting performance (Figure 14) [136], BSA molecularly imprinted alginate composite cryogel membrane was synthesized. The results of the tensile test revealed that the mechanical strength of the obtained membrane has reached 90.00 kPa, and the elongation could reach 93.70%. In addition, the imprinted membrane has a high adsorption capacity of 485.87 mg/g.

Metalloproteinase MMP-1 can be regarded as a disease biomarker, which is meaningful in early diagnosis, so it is of great significance for the selective separation of MMP-1. Epitope peptide was prepared by cleaving MMP-1 in silico with trypsin, and peptide fragments were obtained [137]. The EIPs were synthesized with electropolymerization onto indium tin oxide electrodes and successfully applied for the detection of MMP-1. For the selective extraction of branched cyclodextrins, photo-irradiated MIPs were synthesized using azobenzene as a functional monomer [138], and the purity of cyclodextrin could reach 90.8% after going through MIPs. To reveal the risk of MIPs, the in vivo behavior of nano MIPs was investigated by Kassem and co-workers [139], and nano MIPs were found in each tissue type. The nano MIPs can be cleared via both feces and urine; the low cytotoxicity lays the foundation for in vivo application of nano MIPs. EIPs were synthesized with two-step spin-coating and photopolymerization on polymeric films for the detection of trypsin (Figure 15) [140], and uniform distributed template is achieved. The imprinted film exhibits higher sensitivity of 0.970 and a higher imprinting factor of 4.5.

To solve the problem of lack of effective targeting for fluorescent conjugated polymer (FCP) in biological imaging, sialic acid (SA) was used as a template in the construction of FCP-based MIPs [141]. The obtained SA MIPs showed enhanced fluorescence intensity than NIPs and exhibited selective staining for cancer cells. To modulate the adsorption and release performance of carriers, thermoresponsive EIPs were synthesized with thermal polymerization followed by chemical cross-linking [142]. The obtained EIPs could adsorb 46.6 mg/g of template protein, with an imprinting factor of 4.0. In addition, the template could capture the template at 45 °C and release it at 4 °C. To achieve both precise targeting and drug delivery, dual-template EIPs were fabricated for target diagnosis and drug delivery of pancreatic cancer cells [143]. Modified epitope peptide (Glu-FH) and bleomycin (BLM) were used for the fabrication of dual-template EIPs, and the obtained EIPs not only showed an obvious targeting effect but also showed enhanced inhibiting to cancer cells. A sialic acid imprinted biodegradable nanoparticle-based protein delivery was developed for targeted cancer therapy [144]. With the loading of cytotoxic ribonuclease A (RNase A), the obtained EIPs showed specific tumor-targeting ability and high therapeutic efficacy. PD-L1 peptide imprinted polymers were prepared by precipitation (Figure 16) [145], with incorporation of merocyanine 540 (MC540)-grafted magnetic nanoparticles and green-emitting upconversion nanoparticles. The obtained composites could kill tumor cells precisely, with an enhancing efficacy of photodynamic therapy.

## 6. Conclusions and Future Prospects

Due to the rapid development and application of MIPs, the recognition principle, classification, preparation, and application were summarized for the overall perspective understanding of the MIPs. To expand the application of MIPs and improve the mass transfer rate of templates, surface imprinting materials, and epitope imprinted materials have been rapidly developed based on bulk imprinting materials. The epitope imprinting technology uses exposed peptides as a template, which not only reduces the impact of non-specific adsorption but also avoids the impact of the preparation process on the stability of biological macromolecules. In addition, to solve the issues of long reaction time, high temperature, high energy consumption, and high pollution for traditional preparation techniques, new radiation polymerization based on UV, γ-rays, EB, microwave, ultrasound, plasma, and green polymerization based on scCO_2_, ILs, and DESs have been widely studied and applied. Finally, according to the difference in templates, the latest progress in the recognition applications of MIPs in metal ions, organic molecules, and biomacromolecules was also summarized.

Though the types and preparation methods of imprinting materials have been rapidly developed, and their products have also been widely used, there are still some issues that need to be further investigated.

(1)Intermolecular interaction is the basis for achieving chemical selectivity. However, the current design of interaction between template molecules and functional monomers for ligands is mainly qualitative [12,27,28], which is difficult for the precise construction of imprinting materials. Therefore, quantitative analysis of intermolecular interactions needs to be achieved through computational simulation or other advanced characterization methods.(2)Cavity matching is the foundation for achieving physical selectivity. However, there is limited research on the nanoscale cavity, which can not reveal the influence of intrinsic cavities inside imprinted materials on selectivity. Therefore, it is necessary to optimize the physical selectivity by utilizing PALS, which is sensitive to the determination of the nanoscale [60,64,99,146,147,148].(3)Most of the reported MIPs were only investigated with pure template samples or simulated samples, which may not apply in practice, as the practical samples are more complicated. Therefore, the obtained MIPs should be examined with practical samples.(4)Though the surface imprinting materials and epitope imprinted materials show promising applications, especially for the detection, diagnostics, imaging, and delivery of biomacromolecules, the preparation process is very complicated. For large-scale production and practical application, the synthesis process of these imprinting materials needs optimization.(5)Though the MIPs have an excellent recognition ability, the adsorption capacity is always very low, and future work needs to be focused on the improvement of adsorption capacity.(6)Imprinting polymers for metal cations, organic molecules, and biological macromolecules has been well-developed, but there are few reports about imprinting polymers for the recognition and selective separation of anions [120]. Therefore, it is necessary to develop imprinting polymers of anion and establish the structure-performance relationship.

## Figures and Tables

**Figure 1 polymers-15-02344-f001:**
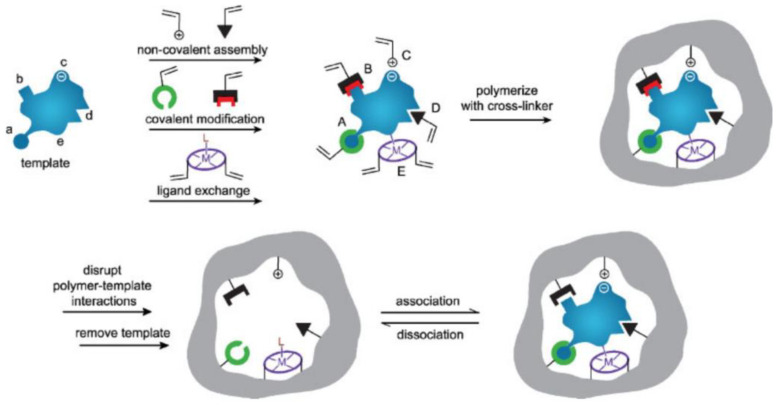
Illustration of the molecular imprinting procedure [27].

**Figure 2 polymers-15-02344-f002:**
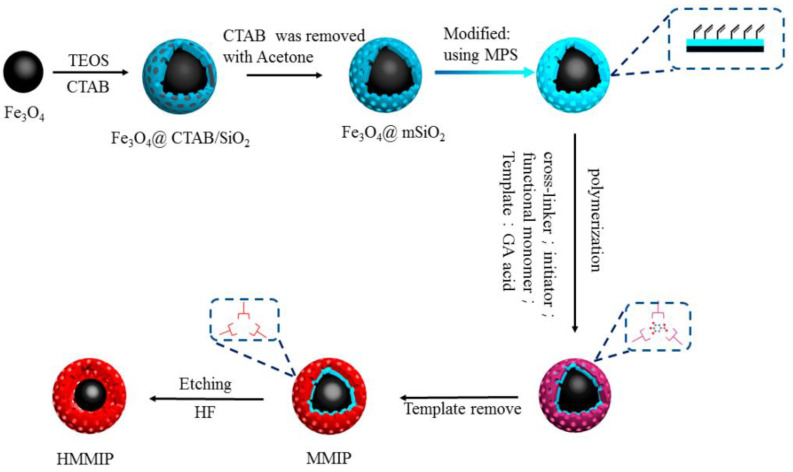
Synthesis of magnetic HMMIP via etching the MMIP [37].

**Figure 3 polymers-15-02344-f003:**
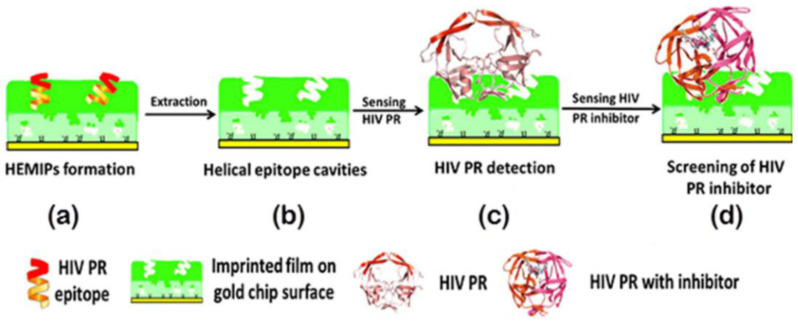
Illustration of the formation of helical epitope-mediated MIPs [39].

**Figure 4 polymers-15-02344-f004:**
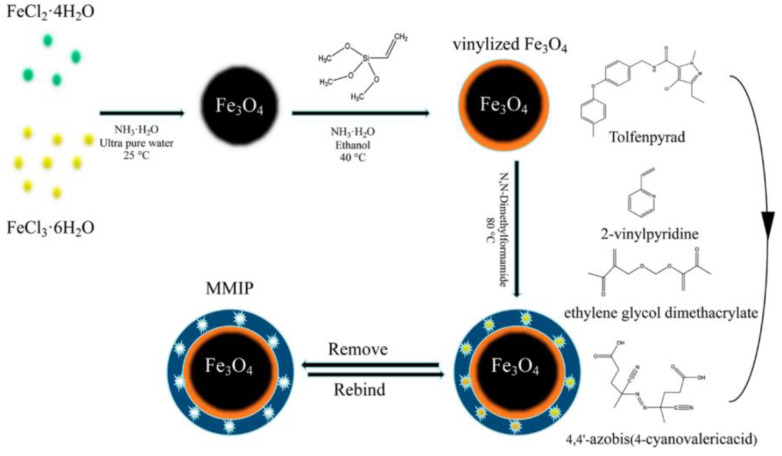
Preparation of MMIP for selective recognition of tolfenpyrad [50].

**Figure 5 polymers-15-02344-f005:**
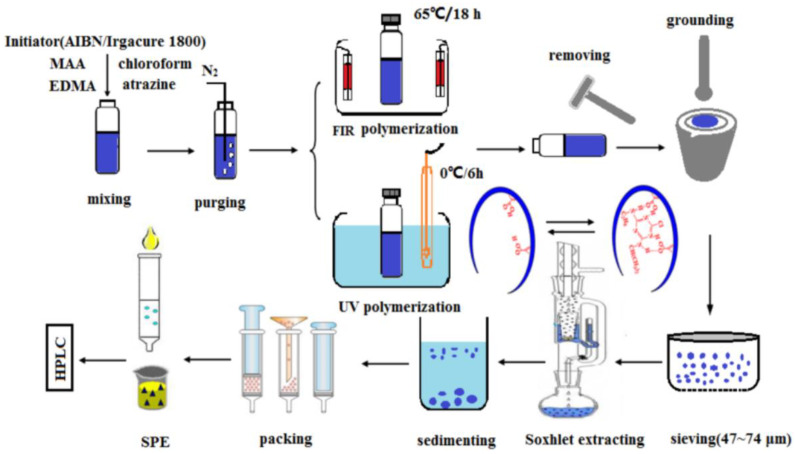
Schematic representative of preparation of atrazine MIPs [58].

**Figure 6 polymers-15-02344-f006:**
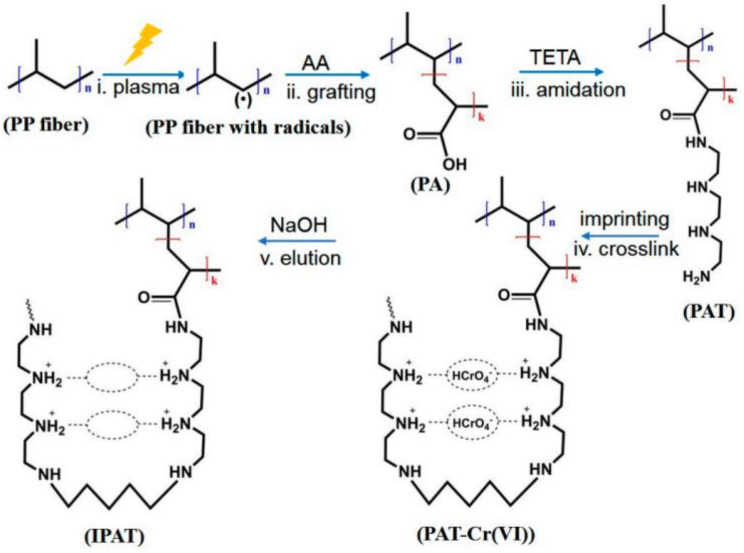
Illustration of the preparation of the ion imprinted fiber [78]. (PP: polypropylene, AA: acrylic acid, TETA: triethylene tetramine, PAT: amide fiber, IPAT: ion-imprinted fiber).

**Figure 7 polymers-15-02344-f007:**
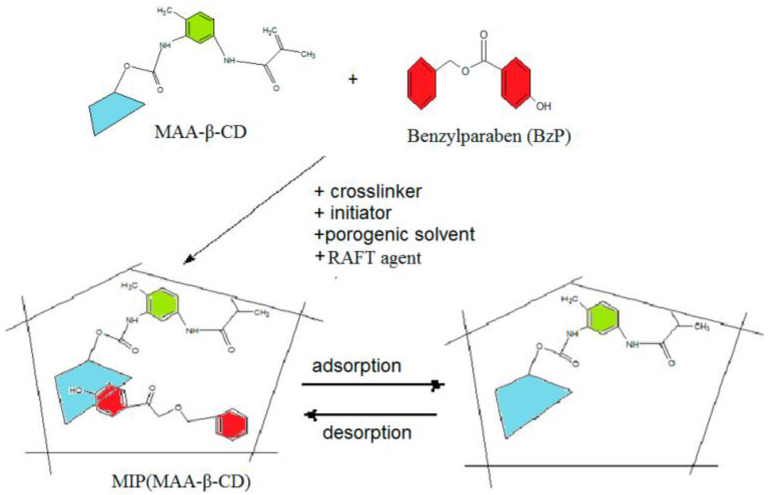
Preparation of MIP-methacrylic acid functionalized β-cyclodextrin [83].

**Figure 8 polymers-15-02344-f008:**
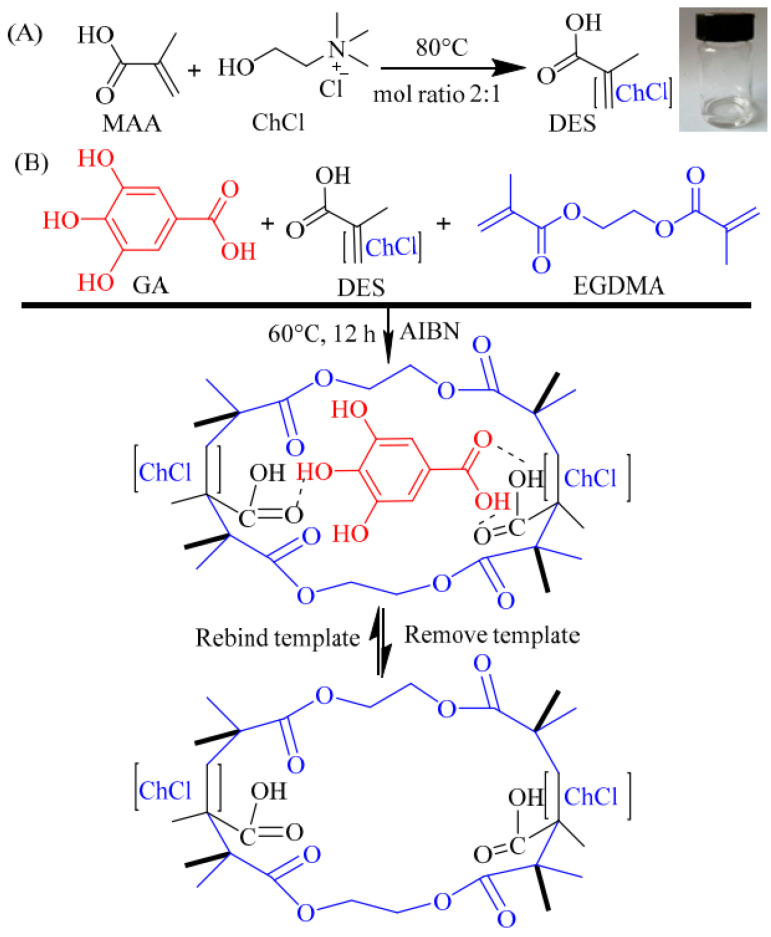
Illustration of the preparation of DESs (**A**) and MIPs (**B**) [99]. (MMA: methylacrylic acid, ChCl: choline chloride, GA: gallic acid, EGDMA: ethylene glycol dimethacrylate).

**Figure 9 polymers-15-02344-f009:**
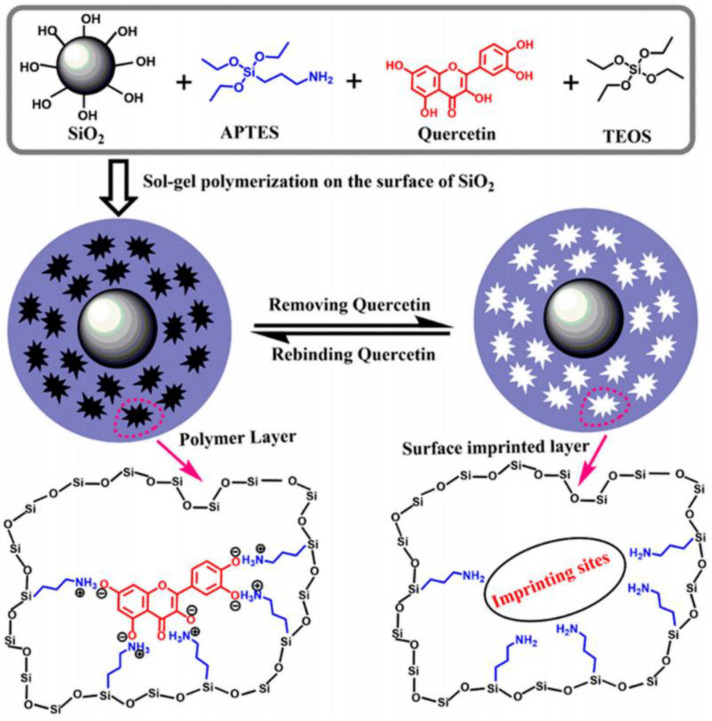
Schematic procedure of MIP preparation for quercetin [101].

**Figure 10 polymers-15-02344-f010:**
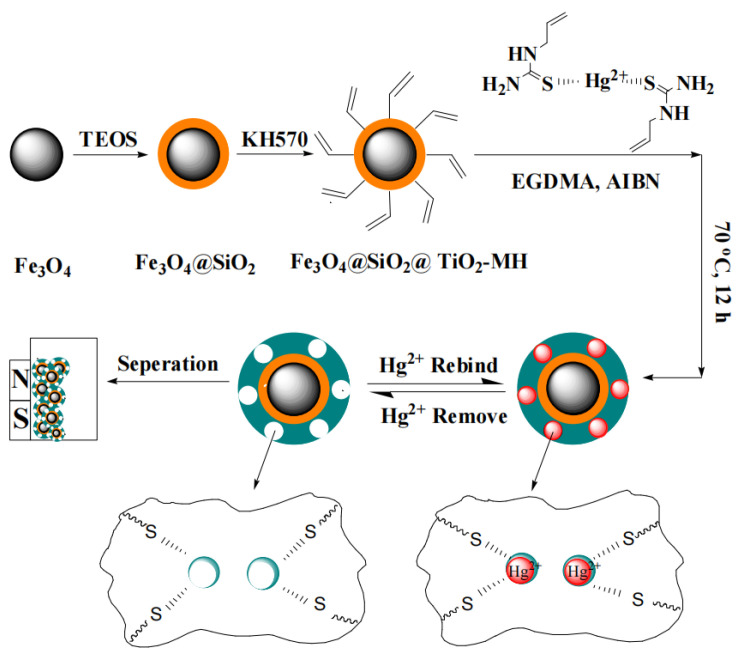
Procedure for the synthesis of magnetic Hg(II)-SIPs [115].

**Figure 11 polymers-15-02344-f011:**
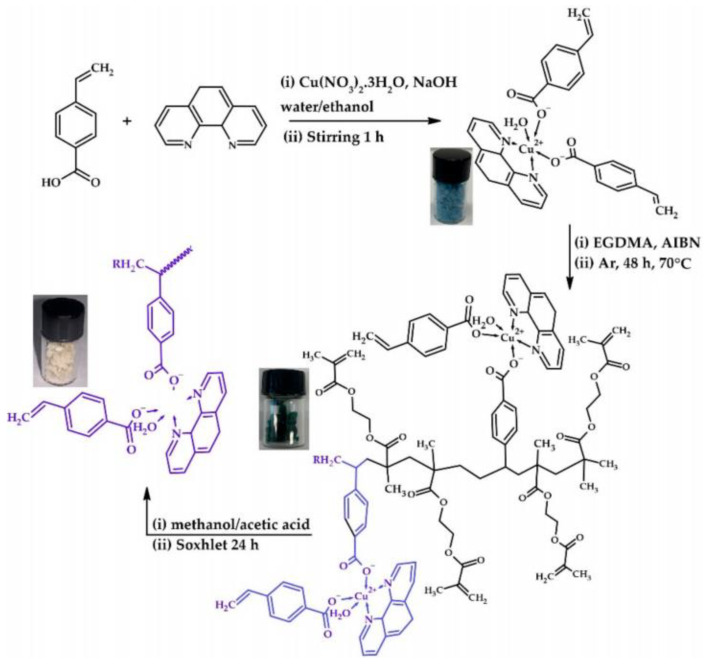
Preparation process of Cu(II)-BIPs [120].

**Figure 12 polymers-15-02344-f012:**
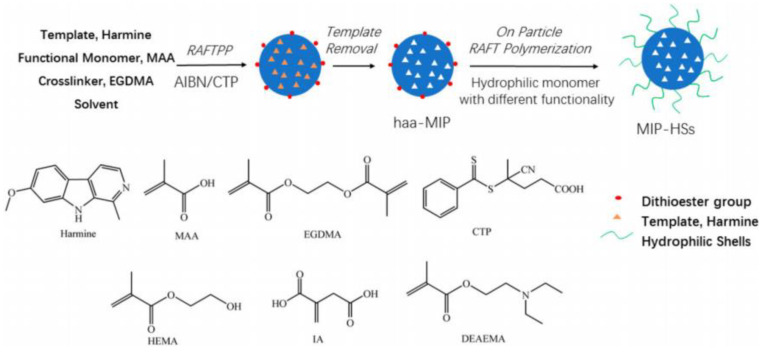
The preparation procedure of the haa-MIPs and structure of MIP-HSs [124].

**Figure 13 polymers-15-02344-f013:**
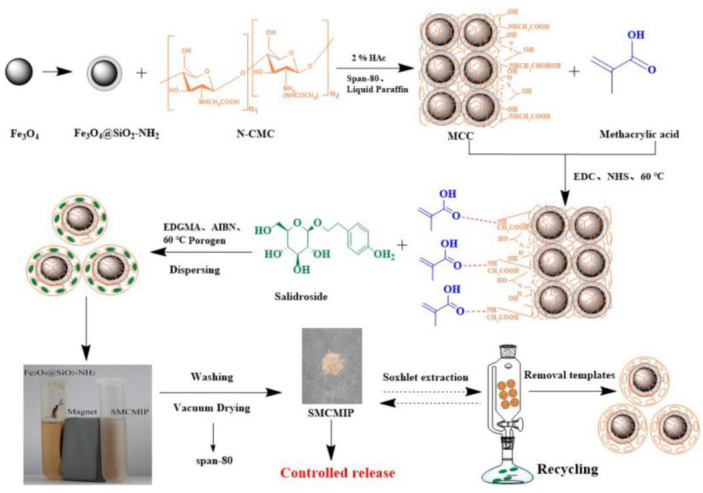
Scheme representing the procedure for preparing salidroside MIPs [128].

**Figure 14 polymers-15-02344-f014:**
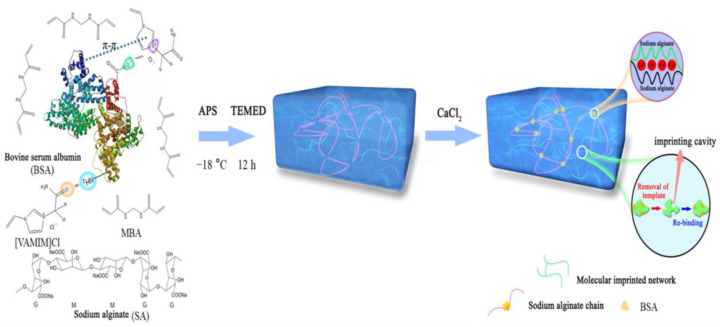
Preparation process of the BSA imprinted membrane [136].

**Figure 15 polymers-15-02344-f015:**
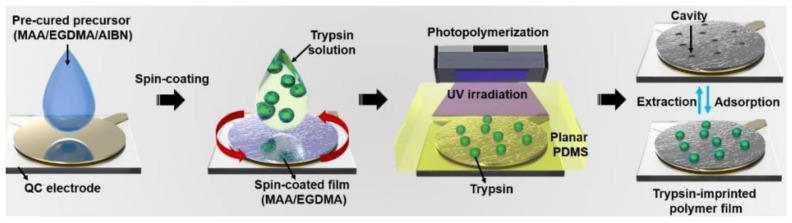
Schematic diagram of the trypsin MIPs fabrication [140].

**Figure 16 polymers-15-02344-f016:**
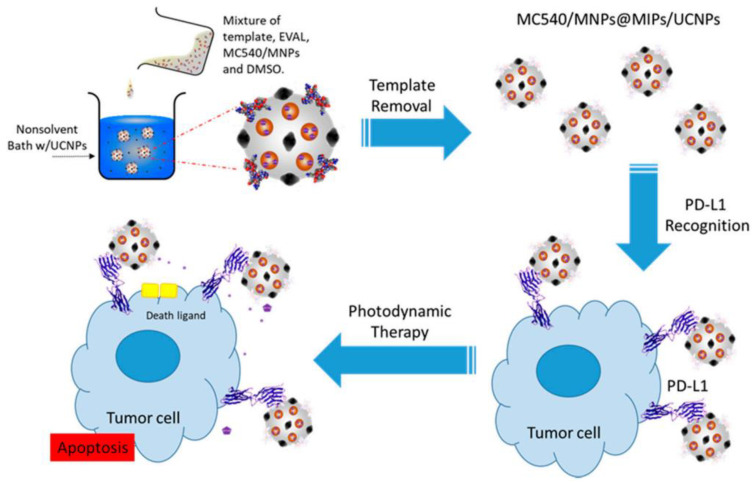
Synthesis of PD-L1 peptide-imprinted composite [145].

**Table 1 polymers-15-02344-t001:** Classification of MIPs.

Type	Preparation Method	Template	Carrier	Application	Ref.
Bulk imprinted polymers	bulk polymerization	Rb^+^	none	decontamination	[34]
	bulk polymerization	melamine	none	sensor	[35]
Surface imprinted polymers	grafting polymerization	BSA	polymeric nanoparticles	sample preparation	[36]
	grafting polymerization	gallic acid	Fe_3_O_4_	sample preparation	[37]
Epitope-imprinted polymers	electrochemical polymerization	ncovS1	Au-TFME	diagnostics	[38]
	irradiated polymerization	peptide	gold chip	therapy	[39]

**Table 2 polymers-15-02344-t002:** MIPs preparation with thermal polymerization.

Thermal Polymerization	Approach	Template	Carrier	Application	Ref.
Traditional	oil bath	benzylpiperazine	none	sample preparation	[49]
	oil bath	tolfenpyrad	Fe_3_O_4_	sample preparation	[50]
Other	oven	quercetin	Fe_3_O_4_	sample preparation	[51]
	magnetic field	nitrophenol	Fe_3_O_4_	decontamination	[52]

**Table 3 polymers-15-02344-t003:** MIPs preparation with radiation polymerization.

Radiation Polymerization	Approach	Template	Carrier	Application	Ref.
UV	solution polymerization	benzyl mercaptan	none	sensor	[54]
	solution polymerization	glutathione	none	sample preparation	[55]
	grafting polymerization	Penicillin G	Fe_3_O_4_	decontamination	[56]
	grafting polymerization	caffeic acid	TiO_2_	sensor	[57]
	solution polymerization	atrazine	none	decontamination	[58]
γ-rays	grafting polymerization	atrazine	fiber	decontamination	[59]
	grafting polymerization	erythromycin	fabrics	sample preparation	[60]
	solution polymerization	phenytoin	none	sample preparation	[61]
	grafting polymerization	bacitracin	membrane	sample preparation	[62]
	solution polymerization	glucose	none	sample preparation	[63]
	bulk polymerization	Er^3+^	none	decontamination	[64]
	solution polymerization	steroid	none	sample preparation	[65]
Electron Beam	solution polymerization	baicalin	none	sample preparation	[66]
	grafting polymerization	ibuprofen	membrane	sample preparation	[67]
	solution polymerization	chloramphenicol	none	sample preparation	[68]
	solution polymerization	quercetin-nickel	none	sample preparation	[69]
	solution polymerization	sulfamethazine	none	sample preparation	[70,71]
	grafting polymerization	Th^3+^	fabrics	decontamination	[72]
Other	microwave radiation	olivetol	none	sample preparation	[73]
	microwave radiation	dimethyl phthalate	none	sample preparation	[74]
	microwave radiation	bisphenol A	none	sample preparation	[75]
	microwave radiation	gibberellin acid	Fe_3_O_4_	sample preparation	[76]
	microwave radiation	quercetin	silica	sample preparation	[77]
	Plasma grafting	Cr^6+^	fiber	decontamination	[78]
	ultrasonic irradiation	naphthol	attapulgite	sample preparation	[79]
	ultrasonic irradiation	caffeine	none	sample preparation	[80]
	ultrasonic irradiation	caffeine	none	sample preparation	[81]

**Table 4 polymers-15-02344-t004:** MIPs preparation with radiation polymerization.

Green Polymerization	Template	Carrier	Application	Ref.
Bulk	sulpiride	none	sample preparation	[82]
	benzylparaben	none	decontamination	[83]
	basic blue	none	decontamination	[84]
	bisphenol A	none	decontamination	[85]
	levofloxacin	none	sample preparation	[86]
Supercritical carbon dioxide	acetamide	none	sample preparation	[87]
	acetamide	none	sample preparation	[88]
	gallic acid	none	sample preparation	[89]
	labdanolic acid	none	sample preparation	[90]
	flufenamic acid	none	drug delivery	[91]
Ionic liquids	isoquercitrin	none	sample preparation	[92]
	naproxon	POSS	sample preparation	[93]
	methyl gallate	none	sample preparation	[94]
	propranolol	none	sample preparation	[95]
	aconitic acid	none	sample preparation	[96]
Deep eutectic solvent	chlorogenic acid	none	sample preparation	[97]
	bovine hemoglobin	none	sample preparation	[98]
	gallic acid	none	sample preparation	[99]
Sol-Gel	gossypol	silica	sample preparation	[29]
	glyphosate	paper	sensor	[100]
	Quercetin	silica	sample preparation	[101]

**Table 5 polymers-15-02344-t005:** Applications of IIPs.

Template	Type	Preparation Method	Liner Range	LOD	Application	Ref.
Pb^2+^	SIPs	thermal polymerization	Max_capacity_ = 51.2 mg/g		decontamination	[111]
Cd^2+^	SIPs	sol–gel	Max_capacity_ = 26.1 mg/g		decontamination	[112]
Cr_2_O_7_^2-^	SIPs	thermal polymerization	Max_capacity_ = 201.55 mg/g		decontamination	[113]
As^3+^	SIPs	sol–gel	2.5–20 μg/L	1.60 μg/L	sample preparation	[114]
Hg^2+^	SIPs	thermal polymerization	Max_capacity_ = 78.3 mg/g		decontamination	[115]
Ni^2+^	BIPs	thermal polymerization	Max_capacity_ = 86.3 mg/g		decontamination	[116]
Zn^2+^	BIPs	thermal polymerization	25–200 μg/L	2.90 μg/L	sample preparation	[117]
Pd^2+^	BIPs	thermal polymerization	Max_capacity_ = 5.085 mg/g		decontamination	[118]
Br^-^	BIPs	chemical cross-linking	Max_capacity_ = 18.89 mg/g		decontamination	[119]
Cu^2+^	BIPs	thermal polymerization	Max_capacity_ = 287.45 mg/g		decontamination	[120]

**Table 6 polymers-15-02344-t006:** Applications of OMIPs.

Template	Type	Preparation Method	Liner Range	LOD	Application	Ref.
Aniline	SIPs	thermal polymerization	1–200 ng/mL	1.0 ng/mL	sample preparation	[121]
Melamine	BIPs	thermal polymerization	6.02–90 μM	6.02 μM	sample preparation	[122]
Methotrexate	SIPs	sol-gel	0.05–250 μg/L	12.51 ng/mL	sample preparation	[123]
Harmine	BIPs	thermal polymerization	Max_capacity_ = 6.0 mg/g		decontamination	[124]
DEHP	SIPs	thermal polymerization	3–2000 μg/L	0.92 ng/mL	sample preparation	[125]
DYKD	EIPs	thermal polymerization	Max_recovery_ = 79.1%		sample preparation	[126]
Naproxen	SIPs	thermal polymerization	S_Naproxen_/R_Naproxen_ = 4.1		enantioseparation	[127]
Salidroside	SIPs	thermal polymerization	total release = 86%		drug delivery	[128]

**Table 7 polymers-15-02344-t007:** Applications of BMIPs.

Template	Type	Preparation Method	Liner Range	LOD	Application	Ref.
Peptide	EIPs	thermal polymerization	Dissociation Constant = 16.8		diagnostics	[134]
Cyt c	SIPs	sol–gel and cross-linking	Max_capacity_ = 86.47 mg/g		sample preparation	[135]
BSA	BIPs	thermal and cross-linking	Max_capacity_ = 485.87 mg/g		sample preparation	[136]
Peptide	EIPs	electropolymerization	0.001 to 10.0 pg/mL	0.2 fg/mL	sample preparation	[137]
Cyclodextrins	BIPs	thermal polymerization	Max_capacity_ = 7.93 μmol/g		sample preparation	[138]
Trypsin	EIPs	thermal polymerization	/		risk assessing	[139]
Trypsin	SIPs	UV radiation	0.006–0.24 μg/mL	25.33 ng/mL	sensor	[140]
Sialic acid	EIPs	chemical cross-linking	enhance fluorescence		biological imaging	[141]
Protein	EIPs	thermal and cross-linking	Max_capacity_ = 46.6 mg/g		sample preparation	[142]
Glu-FH & BLM	EIPs	sol–gel and cross-linking	enhanced inhibiting		targeted therapy	[143]
RNase A	EIPs	sol–gel and cross-linking	high therapeutic efficacy		targeted therapy	[144]
PD-L1	EIPs	grafting and cross-linking	enhancing efficacy of therapy		drug delivery	[145]

## Data Availability

Not applicable.

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
