# Peer review of "Preparation and Application Progress of Imprinted Polymers"

_polymers, 2023, doi:10.3390/polym15102344_

Round 1
Reviewer 1 Report
The review in its current form may not be accepted for publication. Essentially, there seems to be an unpopular categorization of some of the terminilogies of the field under consideration. Also, the organization of the work needs some improvement. However, I believe the manuscript may deserve consideration if the authors are ready to put in the work to revise it based on the issues raised below.
1. English language requires revision if the manuscript is to be well comprehended.
2. Line 28, the 'and' should be removed to aid proper comprehension.
3. Line 30, 'cavity structure' and 'binding site' as used in the same sentence seems confusing.
4. The last sentence of the introduction opens rooms for the question about past reviews on the same topic. Authors should notify and cite appropriately other existing reviews focusing on the same subject.
5. About references:
a. The Reference number 10 (line 62) makes no mention of the classification of MIPs as stated in this manuscript.
b. Authors should check the list of references for consistency with their mention in the text.
c. Some of the citations include several MSc thesis. This seems to be below optimal for a scientific review in an international journal.
d. There are also many references in Chinese that are difficult to find using common sceintific search engines. These are not necessary and should be removed.
6. The recognition principle in section 2 would not in any way be beneficial to readers and could also be misleading. It could be better if authors combine a better form of this with section 3.
7. MIP classification as shown in the first paragraph of section 3 is confusing or rather unpopular.
8. The differentiation between surface imprinting and epitope imprinting as enunciated in the manuscript could be misleading. For instance surface imprinting had been well studied by the group of Syritski using epitopes of target protein. A recent puplication of theirs (https://doi.org/10.1016/j.snb.2021.131160), which should be cited, was in the diagnosis of COVID-19 antigen.
9. Again, the formulation and categorization under section 4 is confusing.
10. In section 5, it could be more beneficial if the headings are adjusted in such a way that each MIP application field examines under it several preparation methodologies. This will better enhance the organization of the manuscript.
A very thorough English language revision is required.
Author Response
The review in its current form may not be accepted for publication. Essentially, there seems to be an unpopular categorization of some of the terminilogies of the field under consideration. Also, the organization of the work needs some improvement. However, I believe the manuscript may deserve consideration if the authors are ready to put in the work to revise it based on the issues raised below.
- English language requires revision if the manuscript is to be well comprehended.
Response: Thanks for the reviewer’s valuable suggestions, the English language of the whole manuscript was revised by a colleague with overseas study experience, and the whole manuscript was revised to our best, we hope the revision will meet the requirement for publication.
- Line 28, the 'and' should be removed to aid proper comprehension.
Response: Thanks for the reviewer’s valuable suggestions, the ‘and’ in line 28 was removed.
- Line 30, 'cavity structure' and 'binding site' as used in the same sentence seems confusing.
Response: Thanks for the reviewer’s valuable suggestions, the whole sentence was revised.
In the revised manuscript:
...which can completely match template in cavity structure, and possess specific recognition functional groups/binding sites for template molecules.
- The last sentence of the introduction opens rooms for the question about past reviews on the same topic. Authors should notify and cite appropriately other existing reviews focusing on the same subject.
Response: Thanks for the reviewer’s valuable suggestions, other existing reviews focusing on the same subject were also cited.
In the revised manuscript:
Though many reviews of MIPs about specific aspects have been reported, there are few reviews about the classification, preparation, and application [13-19]. As the rapid development and application of new MIPs, now is an appropriate time to summarize the recent progress [20-26].
Ref.
- Song, Z.H.; Li, J.H.; Lu, W.H.; Li, B.W.; Yang, G.Q.; Bi, Y.; Arabi, M.; Wang, X.Y.; Ma, J.P.; Chen, L.X. Molecularly imprinted polymers based materials and their applications in chromatographic and electrophoretic separations. Trends Anal. Chem., 2022, 146, 116504. DOI: 1016/j.trac.2021.116504.
- Moein, M.M. Advancements of chiral molecularly imprinted polymers in separation and sensor fields: A review of the last decade. Talanta, 2021, 224, 121794. DOI: 1016/j.talanta.2020.121794.
- Rutkowska, M.; Wasylka, J.P.; Morrison, C.; Wieczorek, P.P.; Namieśnik, J.; Marć, M. Application of molecularly imprinted polymers in an analytical chiral separation and analysis. Trends Anal. Chem., 2018, 102, 91-102. DOI: 1016/j.trac.2018.01.011.
- Ramanavicius, S.; Bubniene, U.S.; Ratautaite, V.; Bechelany, M.; Ramanavicius, A. Electrochemical molecularly imprinted polymer based sensors for pharmaceutical and biomedical applications. Pharm. Biomed. Anal., 2022, 215, 114739. DOI: 10.1016/j.jpba.2022.114739f.
- Wang, J.H.; Liang, R.M.; Qin, W. Molecularly imprinted polymer-based potentiometric sensors. Trends Anal. Chem., 2020, 130, 115980. DOI: 1016/j.trac.2020.115980.
- Ansari, S.; Masounm, S. Recent advances and future trends on molecularly imprinted polymer-based fluorescence sensors with luminescent carbon dots. Talanta, 2021, 223, 121411. DOI: 1016/j.talanta.2020.121411.
- Ahmad, O.S.; Bedwell, T.S.; Esen, C.; Cruz, A.G.; Piletsky, S.A. Molecularly imprinted polymers in electrochemical and optical sensors. Trends Biotechnol., 2019, 37, 294-309. DOI: 1016/j.tibtech.2018.08.009.
- Jancuzura, M.; Luliński, P.; Sobiech, M. Imprinting technology for effective sorbent fabrication: Current state-of-art and future prospects. Materials, 2021, 14, 1850. DOI: 3390/ma14081850.
- Parisi, O.I.; Francomano, F.; Dattilo, M.; Patitucci, F.; Prete, S.; Amone, F.; Puoci, F. The evolution of molecular recognition: From antibodies to molecularly imprinted polymers (MIPs) as artificial counterpart. Funct. Biomater., 2022, 13, 12. DOI: 10.3390/jfb13010012.
- Reville, E.K.; Sylvester, E.H.; Benware, S.J.; Negi, S.S.; Berda, E.B. Customizable molecular recognition: advancements in design, synthesis, and application of molecularly imprinted polymers. Chem., 2022, 13, 3387-3411. DOI: : 10.1039/d1py01472b.
- Cui, B.C.; Liu, P.; Liu, X.J.; Liu, S.Z.; Zhang, Z.H. Molecularly imprinted polymers for electrochemical detection and analysis: progress and perspectives. Mater. Res. Technol., 2020, 9, 12568-12584. DOI: 10.1016/j.jmrt.2020.08.052.
- Zhou, T.Y.; Ding, L.; Che, G.B.; Jiang, W.; Sang, L. Recent advances and trends of molecularly imprinted polymers for specifific recognition in aqueous matrix: Preparation and application in sample pretreatment. Trends Anal. Chem., 2019, 114, 11-28. DOI: 1016/j.trac.2019.02.028.
- Dabrowski, M.; Lach, P.; Cieplak, M.; Kutner, W. Nanostructured molecularly imprinted polymers for protein chemosensing. Bioelectron., 2018, 102, 17-26. DOI: 10.1016/j.bios.2017.10.045.
- Budnicka, M.; Sobiech, M.; Kolmsa, J.; Luliński, P. Frontiers in ion imprinting of alkali- and alkaline-earth metal ions e Recent advancements and application to environmental, food and biomedical analysis. Trends Anal. Chem., 2022, 156, 116711. DOI: 1016/j.trac.2022.116711.
- About references:
- The Reference number 10 (line 62) makes no mention of the classification of MIPs as stated in this manuscript.
Response: Thanks for the reviewer’s valuable suggestions. The classification of MIPs was summarized and originated from Ref. 12, and the Ref. number 10 has been revised to 12.
- Authors should check the list of references for consistency with their mention in the text.
Response: Thanks for the reviewer’s valuable suggestions, the consistency of references has been checked carefully for the whole manuscript.
- Some of the citations include several MSc thesis. This seems to be below optimal for a scientific review in an international journal.
Response: Thanks for the reviewer’s valuable suggestions, all of the three MSc thesis (Ref. 36,41,42 in the revised manuscript) have been replaced with articles from international journals.
Ref.
36 Zhang, Y.; Zhao, G.L.; Han, K.Y.; Sun, D.N.; Zhou, N.; Song, Z.H.; Liu, H.T.; Li, J.H.; Li, G.S. Applications of molecular imprinting technology in the study of traditional chinese medicine. Molecules, 2023, 28, 301. DOI: 10.3390/molecules28010301.
41 Afzal, A.; Mujahid, A.; Schirhagl, R.; Bajwa, S.Z.; Latif, U.; Feroz, S. Gravimetric viral diagnostics: QCM based biosensors for early detection of viruses. Chemosensors, 2017, 5, 7. DOI: 10.3390/chemosensors5010007.
42 Ertürk, G.; Mattiasson, B. Molecular imprinting techniques used for the preparation of biosensors. Sensors, 2017, 17, 288. DOI: 10.3390/s17020288.
- There are also many references in Chinese that are difficult to find using common sceintific search engines. These are not necessary and should be removed.
Response: Thanks for the reviewer’s valuable suggestions, review articles in Chinese (Ref.4,6, 30,32 in the revised manuscript) has been replaced with articles from international journals. As there are few reports about the synthesis of MIPs with Electron Beam Radiation polymerization in English, so the references about the synthesis of MIPs with EBR in Chinese (Ref.47,67-70 in the revised manuscript) has been reserved.
Ref.
4 Ali, G.K.; Omer, K.M. Molecular imprinted polymer combined with aptamer (MIP-aptamer) as a hybrid dual recognition element for bio(chemical) sensing applications. Review. Talanta, 2022, 236, 122878. DOI: 10.1016/j.talanta.2021.122878.
6 Hasanah, A.N.; Safitri, N.; Zulfa, A.; Neli, N.; Rahayu, D. Factors affecting preparation of molecularly imprinted polymer and methods on finding template-monomer interaction as the key of selective properties of the materials. Molecules, 2021, 26, 5612. DOI: 10.3390/molecules26185612.
30 Akgönüllü, S.; Kiliç, S.; Esen, C.; Denizli, A. Molecularly imprinted polymer-based sensors for protein detection. Polymers, 2023, 15, 629. DOI: 10.3390/polym15030629.
32 Ayivi, R.D.; Adesanmi, B.O.; McLamore, E.S.; Wei, J.J.; Obare, S.O. Molecularly imprinted plasmonic sensors as nano-transducers: An effective approach for environmental monitoring applications. Chemosensors, 2023, 11, 203. DOI: 10.3390/chemosensors11030203.
- The recognition principle in section 2 would not in any way be beneficial to readers and could also be misleading. It could be better if authors combine a better form of this with section 3.
Response: Thanks for the reviewer’s valuable suggestions, the recognition principle of MIPs was summarized and originated from Ref. 11,12,27,28 in the revised manuscript. As the recognition principle is very important to the understand of MIP, so “2. Recognition Principles of Imprinted Polymers’ listed alone.
- MIP classification as shown in the first paragraph of section 3 is confusing or rather unpopular.
Response: Thanks for the reviewer’s valuable suggestions. Generally, the MIPs classified with the application fields such as preparation, analysis and detection, environment protection, etc., which is not benefit for the researchers come from different areas to understand the overall perspective of MIPs, as well as the rationable design of MIP structure. Thus the MIPs were classified to bulk imprinted polymers, surface imprinted polymers, and epitope imprinted polymers with the difference of structure, based on extensive literature research.
- The differentiation between surface imprinting and epitope imprinting as enunciated in the manuscript could be misleading. For instance surface imprinting had been well studied by the group of Syritski using epitopes of target protein. A recent puplication of theirs (https://doi.org/10.1016/j.snb.2021.131160), which should be cited, was in the diagnosis of COVID-19 antigen.
Response: Thanks for the reviewer’s valuable suggestions, the definition of surface imprinting and epitope imprinting was adopted from corresponding references, as well as the preparation processes of MIPs based on surface imprinting and epitope imprinting. The publication of Epitope-Imprinted Polymers used in the diagnosis of COVID-19 antigen has been cited.
In the revised manuscript:
For the simple and early detection of COVID-19, EIPs based on SARS-CoV-2 spike protein subunit S1 (ncovS1) was fabricated and applied in an electrochemical sensor [45]. The obtained sensor showed a short reaction time of 15 min, and can detect ncovS1 both in buffer solution and patient’s nasopharyngeal.
Ref.
45 Ayankojo, A. G.; Boroznjak, R.; Reut, J.; Öpik, A.; Syritski, V. Molecularly imprinted polymer based electrochemical sensor for quantitative detection of SARS-CoV-2 spike protein. Sens. Actuators B Chem., 2022, 353, 131160. DOI: 10.1016/j.snb.2021.131160.
- Again, the formulation and categorization under section 4 is confusing.
Response: Thanks for the reviewer’s valuable suggestions. Section 4 is the preparation progress of imprinted polymers, as we all know the MIPs can synthesize with traditional thermal polymerization and high efficient radiation polymerization. To solve the environmental pollution, recent years the green polymerization technology developed rapidly. Thus the section of preparation progress of imprinted polymers was organized as thermal polymerization, radiation polymerization, and green polymerization.
- In section 5, it could be more beneficial if the headings are adjusted in such a way that each MIP application field examines under it several preparation methodologies. This will better enhance the organization of the manuscript.
Response: Thanks for the reviewer’s valuable suggestions, the preparation methods and performance in each MIP application was summarized in table 5,6,7.
In the revised manuscript:
Table 5. Applications of IIPs.
|
Template |
Type |
Preparation method |
Liner range |
LOD |
Application |
Ref. |
|
Pb2+ |
SIPs |
thermal polymerization |
Maxcapacity = 51.2 mg/g |
decontamination |
[111] |
|
|
Cd2+ |
SIPs |
sol-gel |
Maxcapacity = 26.1 mg/g |
decontamination |
[112] |
|
|
Cr2O72- |
SIPs |
thermal polymerization |
Maxcapacity 201.55 mg/g |
decontamination |
[113] |
|
|
As3+ |
SIPs |
sol-gel |
2.5-20 μg/L |
1.60 μg/L |
sample preparation |
[114] |
|
Hg2+ |
SIPs |
thermal polymerization |
Maxcapacity = 78.3 mg/g |
decontamination |
[115] |
|
|
Ni2+ |
BIPs |
thermal polymerization |
Maxcapacity = 86.3 mg/g |
decontamination |
[116] |
|
|
Zn2+ |
BIPs |
thermal polymerization |
25-200 μg/L |
2.90 μg/L |
sample preparation |
[117] |
|
Pd2+ |
BIPs |
thermal polymerization |
Maxcapacity = 5.085 mg/g |
decontamination |
[118] |
|
|
Br- |
BIPs |
chemical cross-linking |
Maxcapacity = 18.89 mg/g |
decontamination |
[119] |
|
|
Cu2+ |
BIPs |
thermal polymerization |
Maxcapacity = 287.45 mg/g |
decontamination |
[120] |
|
Table 6. Applications of OMIPs.
|
Template |
Type |
Preparation method |
Liner range |
LOD |
Application |
Ref. |
|
Aniline |
SIPs |
thermal polymerization |
1-200 ng/mL |
1.0 ng/mL |
sample preparation |
[121] |
|
Melamine |
BIPs |
thermal polymerization |
6.02-90 μM |
6.02 μM |
sample preparation |
[122] |
|
Methotrexate |
SIPs |
sol-gel |
0.05-250 μg/L |
12.51 ng/mL |
sample preparation |
[123] |
|
Harmine |
BIPs |
thermal polymerization |
Maxcapacity = 6.0 mg/g |
decontamination |
[124] |
|
|
DEHP |
SIPs |
thermal polymerization |
3-2000 μg/L |
0.92 ng/mL |
sample preparation |
[125] |
|
DYKD |
EIPs |
thermal polymerization |
Maxrecovery = 79.1% |
sample preparation |
[126] |
|
|
Naproxen |
SIPs |
thermal polymerization |
SNaproxen/RNaproxen = 4.1 |
enantioseparation |
[127] |
|
|
Salidroside |
SIPs |
thermal polymerization |
total release = 86% |
drug delivery |
[128] |
|
Table 7. Applications of OMIPs.
|
Template |
Type |
Preparation method |
Liner range |
LOD |
Application |
Ref. |
|
Peptide |
EIPs |
thermal polymerization |
Dissociation Constant = 16.8 |
diagnostics |
[134] |
|
|
Cyt c |
SIPs |
sol-gel & cross-linking |
Maxcapacity = 86.47 mg/g |
sample preparation |
[135] |
|
|
BSA |
BIPs |
thermal & cross-linking |
Maxcapacity = 485.87 mg/g |
sample preparation |
[136] |
|
|
Peptide |
EIPs |
electropolymerization |
0.001 to 10.0 pg/mL |
0.2 fg/mL |
sample preparation |
[137] |
|
Cyclodextrins |
BIPs |
thermal polymerization |
Maxcapacity = 7.93 μmol/g |
sample preparation |
[138] |
|
|
Trypsin |
EIPs |
thermal polymerization |
/ |
risk assessing |
[139] |
|
|
Trypsin |
SIPs |
UV radiation |
0.006-0.24 μg/mL |
25.33 ng/mL |
sensor |
[140] |
|
Sialic acid |
EIPs |
chemical cross-linking |
enhance fluorescence |
biological imaging |
[141] |
|
|
Protein |
EIPs |
thermal & cross-linking |
Maxcapacity = 46.6 mg/g |
sample preparation |
[142] |
|
|
Glu-FH & BLM |
EIPs |
sol-gel & cross-linking |
enhanced inhibiting |
targeted therapy |
[143] |
|
|
RNase A |
EIPs |
sol-gel & cross-linking |
high therapeutic efficacy |
targeted therapy |
[144] |
|
|
PD-L1 |
EIPs |
grafting & cross-linking |
enhancing efficacy of therapy |
drug delivery |
[145] |
|

Reviewer 2 Report
Please see attachment.

Extensive editing of English language is required
Author Response
This review presents the classification, the preparation methods, and the practical applications of the molecularly imprinted polymers.
- The reviewer does not recommend the publication of this review because it lacks novelty and has a limited choice of examples. A review should gather a multitude of examples in a table instead of citing some examples from the bibliography. Furthermore, no new figures or tables were generated for this review, and all figures are reproduced from previously published papers.
Response: Thanks for the reviewer’s valuable suggestions. Firstly, the classification of imprinted polymers (bulk imprinted polymers, surface imprinted polymers, and epitope imprinted polymers), preparation methods (thermal polymerization, radiation polymerization, and green polymerization), and application (metal ions, organic molecules, and biological macromolecules) in this review is quiet different from most reviews about MIP, which classified with the application fields such as sample preparation, analysis and detection, environment protection, etc. The traditional classification is not benefit for the researchers come from different areas to understand the overall perspective of MIPs, thus this review does not lack novelty. Secondly, there are 148 references in the revised manuscript, which is sufficient for a common reviews. Finally, seven tables were added according to the author’s comment.
In the revised manuscript:
Table 1. Classification of MIPs.
|
Type |
Preparation method |
Template |
Carrier |
Application |
Ref. |
|
Bulk imprinted polymers |
bulk polymerization |
Rb+ |
none |
decontamination |
[34] |
|
bulk polymerization |
melamine |
none |
sensor |
[35] |
|
|
Surface imprinted polymers |
grafting polymerization |
BSA |
polymeric nanoparticles |
sample preparation |
[38] |
|
grafting polymerization |
gallic acid |
Fe3O4 |
sample preparation |
[39] |
|
|
Epitope-imprinted polymers |
electrochemical polymerization |
ncovS1 |
Au-TFME |
diagnostics |
[45] |
|
irradiated polymerization |
peptide |
gold chip |
therapy |
[46] |
Table 2. MIPs preparation with thermal polymerization.
|
Thermcal polymerization |
Approach |
Template |
Carrier |
Application |
Ref. |
|
Traditional |
oil bath |
benzylpiperazine |
none |
sample preparation |
[49] |
|
oil bath |
tolfenpyrad |
Fe3O4 |
sample preparation |
[50] |
|
|
Other |
oven |
quercetin |
Fe3O4 |
sample preparation |
[51] |
|
magnetic field |
nitrophenol |
Fe3O4 |
decontamination |
[52] |
Table 3. MIPs preparation with radiation polymerization.
|
Radiation polymerization |
Approach |
Template |
Carrier |
Application |
Ref. |
|
UV |
solution polymerization |
benzyl mercaptan |
none |
sensor |
[54] |
|
solution polymerization |
glutathione |
none |
sample preparation |
[55] |
|
|
grafting polymerization |
Penicillin G |
Fe3O4 |
decontamination |
[56] |
|
|
grafting polymerization |
caffeic acid |
TiO2 |
sensor |
[57] |
|
|
solution polymerization |
atrazine |
none |
decontamination |
[58] |
|
|
γ-rays |
grafting polymerization |
atrazine |
fiber |
decontamination |
[59] |
|
grafting polymerization |
erythromycin |
fabrics |
sample preparation |
[60] |
|
|
solution polymerization |
phenytoin |
none |
sample preparation |
[61] |
|
|
grafting polymerization |
bacitracin |
membrane |
sample preparation |
[62] |
|
|
solution polymerization |
glucose |
none |
sample preparation |
[63] |
|
|
bulk polymerization |
Er3+ |
none |
decontamination |
[64] |
|
|
solution polymerization |
steroid |
none |
sample preparation |
[65] |
|
|
Electron Beam |
solution polymerization |
baicalin |
none |
sample preparation |
[66] |
|
grafting polymerization |
ibuprofen |
membrane |
sample preparation |
[67] |
|
|
solution polymerization |
chloramphenicol |
none |
sample preparation |
[68] |
|
|
solution polymerization |
quercetin-nickel |
none |
sample preparation |
[69] |
|
|
solution polymerization |
sulfamethazine |
none |
sample preparation |
[70,71] |
|
|
grafting polymerization |
Th3+ |
fabrics |
decontamination |
[72] |
|
|
Other |
microwave radiation |
olivetol |
none |
sample preparation |
[73] |
|
microwave radiation |
dimethyl phthalate |
none |
sample preparation |
[74] |
|
|
microwave radiation |
bisphenol A |
none |
sample preparation |
[75] |
|
|
microwave radiation |
gibberellin acid |
Fe3O4 |
sample preparation |
[76] |
|
|
microwave radiation |
quercetin |
silica |
sample preparation |
[77] |
|
|
Plasma grafting |
Cr6+ |
fiber |
decontamination |
[78] |
|
|
ultrasonic irradiation |
naphthol |
attapulgite |
sample preparation |
[79] |
|
|
ultrasonic irradiation |
caffeine |
none |
sample preparation |
[80] |
|
|
ultrasonic irradiation |
caffeine |
none |
sample preparation |
[81] |
Table 4. MIPs preparation with radiation polymerization.
|
Green polymerization |
Template |
Carrier |
Application |
Ref. |
|
Bulk |
sulpiride |
none |
sample preparation |
[83] |
|
benzylparaben |
none |
decontamination |
[84] |
|
|
basic blue |
none |
decontamination |
[85] |
|
|
bisphenol A |
none |
decontamination |
[86] |
|
|
levofloxacin |
none |
sample preparation |
[87] |
|
|
Supercritical carbon dioxide |
acetamide |
none |
sample preparation |
[89] |
|
acetamide |
none |
sample preparation |
[90] |
|
|
gallic acid |
none |
sample preparation |
[91] |
|
|
labdanolic acid |
none |
sample preparation |
[92] |
|
|
flufenamic acid |
none |
drug delivery |
[93] |
|
|
Ionic liquids |
isoquercitrin |
none |
sample preparation |
[95] |
|
naproxon |
POSS |
sample preparation |
[96] |
|
|
methyl gallate |
none |
sample preparation |
[97] |
|
|
propranolol |
none |
sample preparation |
[98] |
|
|
aconitic acid |
none |
sample preparation |
[99] |
|
|
Deep Eutectic Solvent |
chlorogenic acid |
none |
sample preparation |
[101] |
|
bovine hemoglobin |
none |
sample preparation |
[102] |
|
|
gallic acid |
none |
sample preparation |
[103] |
|
|
Sol-Gel |
gossypol |
silica |
sample preparation |
[29] |
|
glyphosate |
paper |
sensor |
[104] |
|
|
Quercetin |
silica |
sample preparation |
[105] |
Table 5. Applications of IIPs.
|
Template |
Type |
Preparation method |
Liner range |
LOD |
Application |
Ref. |
|
Pb2+ |
SIPs |
thermal polymerization |
Maxcapacity = 51.2 mg/g |
decontamination |
[111] |
|
|
Cd2+ |
SIPs |
sol-gel |
Maxcapacity = 26.1 mg/g |
decontamination |
[112] |
|
|
Cr2O72- |
SIPs |
thermal polymerization |
Maxcapacity 201.55 mg/g |
decontamination |
[113] |
|
|
As3+ |
SIPs |
sol-gel |
2.5-20 μg/L |
1.60 μg/L |
sample preparation |
[114] |
|
Hg2+ |
SIPs |
thermal polymerization |
Maxcapacity = 78.3 mg/g |
decontamination |
[115] |
|
|
Ni2+ |
BIPs |
thermal polymerization |
Maxcapacity = 86.3 mg/g |
decontamination |
[116] |
|
|
Zn2+ |
BIPs |
thermal polymerization |
25-200 μg/L |
2.90 μg/L |
sample preparation |
[117] |
|
Pd2+ |
BIPs |
thermal polymerization |
Maxcapacity = 5.085 mg/g |
decontamination |
[118] |
|
|
Br- |
BIPs |
chemical cross-linking |
Maxcapacity = 18.89 mg/g |
decontamination |
[119] |
|
|
Cu2+ |
BIPs |
thermal polymerization |
Maxcapacity = 287.45 mg/g |
decontamination |
[120] |
|
Table 6. Applications of OMIPs.
|
Template |
Type |
Preparation method |
Liner range |
LOD |
Application |
Ref. |
|
Aniline |
SIPs |
thermal polymerization |
1-200 ng/mL |
1.0 ng/mL |
sample preparation |
[121] |
|
Melamine |
BIPs |
thermal polymerization |
6.02-90 μM |
6.02 μM |
sample preparation |
[122] |
|
Methotrexate |
SIPs |
sol-gel |
0.05-250 μg/L |
12.51 ng/mL |
sample preparation |
[123] |
|
Harmine |
BIPs |
thermal polymerization |
Maxcapacity = 6.0 mg/g |
decontamination |
[124] |
|
|
DEHP |
SIPs |
thermal polymerization |
3-2000 μg/L |
0.92 ng/mL |
sample preparation |
[125] |
|
DYKD |
EIPs |
thermal polymerization |
Maxrecovery = 79.1% |
sample preparation |
[126] |
|
|
Naproxen |
SIPs |
thermal polymerization |
SNaproxen/RNaproxen = 4.1 |
enantioseparation |
[127] |
|
|
Salidroside |
SIPs |
thermal polymerization |
total release = 86% |
drug delivery |
[128] |
|
Table 7. Applications of OMIPs.
|
Template |
Type |
Preparation method |
Liner range |
LOD |
Application |
Ref. |
|
Peptide |
EIPs |
thermal polymerization |
Dissociation Constant = 16.8 |
diagnostics |
[134] |
|
|
Cyt c |
SIPs |
sol-gel & cross-linking |
Maxcapacity = 86.47 mg/g |
sample preparation |
[135] |
|
|
BSA |
BIPs |
thermal & cross-linking |
Maxcapacity = 485.87 mg/g |
sample preparation |
[136] |
|
|
Peptide |
EIPs |
electropolymerization |
0.001 to 10.0 pg/mL |
0.2 fg/mL |
sample preparation |
[137] |
|
Cyclodextrins |
BIPs |
thermal polymerization |
Maxcapacity = 7.93 μmol/g |
sample preparation |
[138] |
|
|
Trypsin |
EIPs |
thermal polymerization |
/ |
risk assessing |
[139] |
|
|
Trypsin |
SIPs |
UV radiation |
0.006-0.24 μg/mL |
25.33 ng/mL |
sensor |
[140] |
|
Sialic acid |
EIPs |
chemical cross-linking |
enhance fluorescence |
biological imaging |
[141] |
|
|
Protein |
EIPs |
thermal & cross-linking |
Maxcapacity = 46.6 mg/g |
sample preparation |
[142] |
|
|
Glu-FH & BLM |
EIPs |
sol-gel & cross-linking |
enhanced inhibiting |
targeted therapy |
[143] |
|
|
RNase A |
EIPs |
sol-gel & cross-linking |
high therapeutic efficacy |
targeted therapy |
[144] |
|
|
PD-L1 |
EIPs |
grafting & cross-linking |
enhancing efficacy of therapy |
drug delivery |
[145] |
|
- There are some fundamental mistakes in the classification of sol-gel polymerization under thermal polymerization.
Response: Thanks for the reviewer’s valuable suggestions. As mentioned in the 8th comment of the reviewer, sol-gel polymerization can be regarded as a green solvent synthesis, so the “4.1.2. Sol-Gel Polymerization” under thermal polymerization was moved to the “4.3.5. Sol-Gel Polymerization” under green polymerization.
- The conclusion is inadequate and should contain a summary of the main points covered in the review. Additionally, the conclusion should present suggestions for future research or practical applications. The addressed limitations in the conclusion are too restrictive.
Response: Thanks for the reviewer’s valuable suggestions, the conclusion has been revised according to the reviewer’s comments in the revised manuscript. Firstly, the main points were covered in the first paragraph. Secondly, the limitations of the MIPs was extended, and corresponding suggestions was present for future research or practical applications.
In the revised manuscript:
Due to its rapid development and application of MIPs, the recognition principle, classification, preparation and application were summarized for the overall perspective understanding of the MIPs. To expand the application of MIPs and improve the mass transfer rate of templates, surface imprinting materials, and epitope imprinted materials have been rapidly developed based on bulk imprinting materials. The epitope imprinting technology using exposed peptide as a template, which not only reduce the impact of non-specific adsorption, but also avoids the impact of the preparation process on the stability of biological macromolecules. In addition, to solve the issues of long reaction time, high temperature, high energy consumption, and high pollution for traditional preparation techniques, new radiation polymerization based on UV, γ-rays, EB, microwave, ultrasound, plasma, and green polymerization based on scCO2, ILs, DESs have been widely studied and applied. Finally, according to the difference of templates, the latest progress in the recognition applications of MIPs in metal ions, organic molecules, and biomacromolecules was also summarized.
Though the types and preparation methods of imprinting materials have been rapidly developed, and their products have also been widely used, there are still some issues that need to be further investigated.
- The intermolecular interaction is the basis for achieving chemical selectivity. However, the current design of interaction between template molecules and functional monomers for ligands is mainly qualitative [12,27,28], which is difficult for the precise construction of imprinting materials. Therefore, quantitative analysis of intermolecular interactions needs to be achieved through computational simulation or other advanced characterization methods.
- Cavity matching is the foundation for achieving physical selectivity. However, there is limited research on the nanoscale cavity, which can not reveal the influence of intrinsic cavities inside imprinted materials on selectivity. Therefore, it is necessary to optimize the physical selectivity by utilizing PALS, which is sensitive to the determination of the nanoscale [61,65,100,146-148].
- Most of the reported MIPs was only investigated with pure template sample or simulated sample, which may not applied in practice, as the practical samples are more complicated. Therefore, the obtained MIPs should examined with practical samples.
- Though the surface imprinting materials and epitope imprinted materials shows promising application, especially for the detection, diagnostics, imaging, and delivery of biomacromolecules, yet the preparation process is very complicated. For the large scale production and practical application, the synthesis process of these imprinting materials need optimizing.
- Tough the MIPs have an excellent recognition ability, the adsorption capacity always very low, future work need focused on the improvement of adsorption capacity.
- Imprinting polymers for metal cations, organic molecules, and biological macromolecules have been well-developed, but there are few reports about imprinting polymers for the recognition and selective separation of anions [120]. Therefore, it is necessary to develop imprinting polymers of anion, and establish the structure-performance relationship.
- The applications cited in the text of the review are not in line with the title "Application Progress." Some important MIP applications are missing, such as drug delivery, sample preparation, decontamination, and sensor application.
Response: Thanks for the reviewer’s valuable suggestions. As shown in table 1-7, most applications of MIPs were cited in the revised manuscript, including decontamination, sensor, sample preparation, diagnostics, therapy, drug delivery, imaging, etc.
Other comments from the reviewer are listed below:
- English language needs editing
Response: Thanks for the reviewer’s valuable suggestions, the English language of the whole manuscript was revised by a colleague with overseas study experience, and the whole manuscript was revised to our best, we hope the revision will meet the requirement for publication.
- Line 139. “polymerization and in the of an initiator”. Please correct this sentence. It seems that
it is incomplete.
Response: Thanks for the reviewer’s valuable suggestions, the sentence has been revised.
In the revised manuscript:
The MIPs always prepared with thermal polymerization and in the presence of an initiator, which needs a long reaction time and high energy consumption.
- Paragraph 4.1.2. Sol-Gel polymerization cannot be under the thermal polymerization main paragraph 4.1. Thermal polymerization because the sol-gel polymerization as mentioned by the authors is usually done at room temperature.
Response: Thanks for the reviewer’s valuable suggestions. As mentioned in the 8th comment of the reviewer, sol-gel polymerization can be regarded as a green solvent synthesis, so the “4.1.2. Sol-Gel Polymerization” under thermal polymerization was moved to the “4.3.5. Sol-Gel Polymerization” under green polymerization.
- Moreover, the chemical reactions behind the sol-gel polymerization should be presented. The advantages of the sol-gel in terms of green solvent synthesis, higher specificity compared to the MIP should be discussed.
Response: Thanks for the reviewer’s valuable suggestions. Firstly, the chemical reaction of sol-gel polymerization was present in Figure 9 in the revised manuscript, the APTES and TEOS hydrolysis to produce silanol, and react with the hydroxyl on the silica. Secondly, for the advantages of the sol-gel in terms of green solvent synthesis, the “4.1.2. Sol-Gel Polymerization” under thermal polymerization was moved to the “4.3.5. Sol-Gel Polymerization” under green polymerization. Finally, the characteristics of the MIPs obtained with sol-gel polymerization and bulk polymerization was also compared in the the revised manuscript.
In the revised manuscript:
4.3.5. Sol-Gel Polymerization
In the separation investigation of gossypol, the sol-gel method was applied for the synthesis of SIPs [29]. Typically, gossypol was dissolved in acetone and (3-aminopropyl) triethoxysilane was added to the mixture to obtain self assemble complexes. Then, activated silica carrier,cross-linker tetraethoxysilane, and acetic acid were added. Finally, SIPs were obtained with reaction at room temperature for 24 h. Results revealed that the MIPs obtained with sol-gel polymerization was a desirable sorbent for rapid adsorption of gossypol, and the MIPs obtained with bulk polymerization was suitable for selective recognition of gossypol. A paper-based fluorescent senor targeting glyphosate, integrated with surface imprinting technology was reported by Wang’s group [104]. the SIPs were prepared by sol-gel polymerization, the obtained SIPs exhibited high selectivity for glyphosate. The detection accuracy of the obtained sensor was relative good, with a recovery rate of 92-117% for practical samples.
Figure 9. Schematic procedure of MIP preparation for quercetin [105]. The sol-gel approach was used for the fabrication of quercetin-based SIPs, the adsorption could reached equilibrium within 90 min, with a maximum adsorption capacity of 35.7 mg/g. The mechanism for adsorption isotherm and kinetics of SIPs was proved to obey the Freunflich isotherm modelp and seudo-second-order kinetics model.
Ref.
29 Wang, L.L.; Zhi, K.K.; Zhang, Y.G.; Liu Y.X.; Zhang, L.T.; Yasin, A.; Lin, Q. F. Molecularly imprinted polymers for gossypol via sol-gel, bulk, and surface layer imprinting-A comparative study. Polymers, 2019,11, 602. DOI: 10.3390/polym11040602.
- Line 205. “which can be used for the preparation of MIPs used in the medical area”. Please
add a reference.
Response: Thanks for the reviewer’s valuable suggestions, Ref. 53 was added.
In the revised manuscript:
Moreover, some radiation polymerization does even not require an initiator, which can be used for the preparation of MIPs used in the medical area [53].
Ref.
53 Magaña, H.; Becerra, C.D.; Medina, A.S.; Palomino, K.; Vizcaíno, G.P.; Sarabia, A.O.; Bucio, E.; Bravo, J.M.C. Radiation grafting of a polymeric prodrug onto silicone rubber for potential medical/sSurgical procedures. Polymers, 2020, 12, 1297. DOI: 10.3390/polym12061297.
- Line 222. Please replace “complex materials” by “complex matrices”.
Response: Thanks for the reviewer’s valuable suggestions, “complex materials” was replaced by “complex matrices”.
- Line 225. Please replace AIB by AIBN.
Response: Thanks for the reviewer’s valuable suggestions, “AIB” was replaced by “AIBN”.
- Line 249. What is imprinted fabrics?
Response: Thanks for the reviewer’s valuable suggestions, “fabrics” was replaced by “porous polythylene/polypropylene nonwoven fabrics”.
- Line 253 and 291. What is AM?
Response: Thanks for the reviewer’s valuable suggestions, “AM” was replaced by “acrylamide”.
- Line 313. Please give the adsorption capacity in mol/g of material.
Response: Thanks for the reviewer’s valuable suggestions, the adsorption capacity in mol/g of material is not available from the corresponding Ref. 73.
Ref.
73 Jin, Y.F.; Chen, N.; Liu, R.Q.; Zhang, Y.P.; Bai, L.Y.; Chen, J. Rapid preparation of monolithic molecular imprinted polymer fiber for solid phase microextraction by microwave irradiation. J. Chin. Chem. Soc., 2013, 60, 1043-1049. DOI: 10.1002/jccs. 201200548.
- Figure 7. Please define all the abbreviations: PP, AA, TETA, PAT, IPAT.
Response: Thanks for the reviewer’s valuable suggestions, all of the abbreviations were defined.
In the revised manuscript:
Figure 6. Illustration of the ion imprinted fibers preparation [78]. Chromium ion imprinted polypropylene fibers were synthesized by the plasma-mediated grafting method, and the obtained imprinted materials exhibited excellent selectivity to chromium ion compared with non-imprinted fibers (PP: polypropylene, AA: acrylic acid, TETA: triethylene tetramine, PAT: amide fiber, IPAT: ion-imprinted fiber).
- Line 364. Please remove low-cost because Supercritical CO2 is quite expensive.
Response: Thanks for the reviewer’s valuable suggestions, “low-cost” was removed.
- Figure 9. Please define all the abbreviations in the figure legend.
Response: Thanks for the reviewer’s valuable suggestions, all of the abbreviations were defined.
In the revised manuscript:
Figure 8. Illustration of the preparation of DESs (A) and MIPs (B) [103]. Gallic acid MIPs was synthesized with bulk polymerization using DESs as functional monomers, the obtained MIPs has a mesoporous structure with an average pore diameter of 9.65 nm. The adsorption behavior followed pseudo-second-order kinetic model, with a maximum adsorption capacity of 0.711 mmol/g (MMA: methylacrylic acid, ChCl: choline chloride, GA: gallic acid, EGDMA: ethylene glycol dimethacrylate).
- Line 490. Please correct “wew”
Response: Thanks for the reviewer’s valuable suggestions, “wew” was revised to “was” .
- All the figure legends should be developed. A figure and the legend should be self-standing
and it should be no need to go back to the text to understand the figure.
Response: Thanks for the reviewer’s valuable suggestions, all the figure legends were developed, all of the figures and corresponding legends were combined in the the manuscript.

Reviewer 3 Report
Review: polymers-2363975.
Title: Preparation and application progress of imprinted polymers.
In this review-type manuscript, Authors have discussed various aspects of the synthetic approaches and application possibilities of the imprinted polymers. The enormous increase of original papers devoted to imprinted polymers in recent years requires cumulative or perspective discussion. Moreover, the potential of imprinted polymers, mostly in analytical chemistry attracted attention of many scientific groups. Thus, the review type manuscript could be a proper tool to summarize current state-of-art in the field and to outline future prospects with high potential for Readers. However, the following points should be addressed by Authors at this stage of evaluation:
1/ The novelty should be emphasized together with the explanation, why there is still a room to provide another review type paper, summarizing the field of molecularly or/and ion-imprinted polymers. It should be supported by recent reviews (see: Trends Anal. Chem. 2022, 146, 116504, Talanta 2021, 224, 121794, Trends Anal. Chem. 2018, 102, 91-102, J. Pharm. Biomed. Anal. 2022, 215, 114739, Trends Anal. Chem. 2020, 130, 115980, Talanta 2021, 224, 121794, Talanta 2021, 223 (1), 121411, Trends Biotechnol. 2019, 37, 294-309, Materials 2021, 14, 1850, J. Funct. Biomater. 2022, 13, 12, Polym. Chem. 2022, 13, 3387-3411, J. Mater. Res. Technol. 2020, 9, 12568-12584, Trends Anal. Chem. 2019, 114, 11-28, Biosens. Bioelectron. 2018, 102, 17-26, Trends Anal. Chem. 2022, 156, 116711).
2/ An important aspect to be considered is the fact that tables have not been included. Tables summarizing and highlighting the most important aspects of the articles found in the literature provide valuable and clear information to the Reader. This is extremely necessary in order to provide and demonstrate a suitable and complete revision of the literature. Please, add at least one item to the manuscript, for instance summarizing the application possibilities of imprinted polymers.
3/ In my opinion, the Section 5 devoted to imprinted polymers for biomacromolecules should be extended since it focuses on the very important aspect of application, viz. therapy and diagnostics. Please, extend the discussion and complete the references (see: Acta Biomater. 2020, 101, 444-458, Trends Biotechnol. 2020, 38, 368-387, Theranostics 2022, 12, 2406-2426, Polymers 2019, 11, 2085, Trends Anal. Chem. 2021, 143, 116414, Biomedicines 2021, 9, 1923, Appl. Mater. Interfaces 2017, 9, 3006-3015, ACS Nano 2021, 15, 18214-18225, Appl. Mater. Interfaces 2016, 8, 5747-5751, Appl. Mater. Interfaces 2019, 11, 32431-32440).
In my opinion, selected older references (e.g. Refs. 4 and 12) as well as references in Chinese (e.g. Refs. 4 and 16) could be deleted and/or replaced by new ones, preferably provided in English. Unless it is highly necessary, Authors shall re-think reference to Master Thesis (Refs 27 and 28).
Minor editorial issues: Ref. 51 – should be: Gadzala-Kopciuch, R., Ref. 40 – should be Anal. Lett., Ref. 88 – should be J. Chromatogr. Sci. and similarly Ref. 109.
Based on above, I recommend major revision of the manuscript.
Author Response
In this review-type manuscript, Authors have discussed various aspects of the synthetic approaches and application possibilities of the imprinted polymers. The enormous increase of original papers devoted to imprinted polymers in recent years requires cumulative or perspective discussion. Moreover, the potential of imprinted polymers, mostly in analytical chemistry attracted attention of many scientific groups. Thus, the review type manuscript could be a proper tool to summarize current state-of-art in the field and to outline future prospects with high potential for Readers. However, the following points should be addressed by Authors at this stage of evaluation:
- The novelty should be emphasized together with the explanation, why there is still a room to provide another review type paper, summarizing the field of molecularly or/and ion-imprinted polymers. It should be supported by recent reviews (see: Trends Anal. Chem. 2022, 146, 116504, Talanta 2021, 224, 121794, Trends Anal. Chem. 2018, 102, 91-102, J. Pharm. Biomed. Anal. 2022, 215, 114739, Trends Anal. Chem. 2020, 130, 115980, Talanta 2021, 224, 121794,Talanta 2021, 223 (1), 121411, Trends Biotechnol. 2019, 37, 294-309, Materials 2021, 14, 1850, J. Funct. Biomater. 2022, 13, 12, Polym. Chem. 2022, 13, 3387-3411, J. Mater. Res. Technol. 2020, 9, 12568-12584, Trends Anal. Chem. 2019, 114, 11-28, Biosens. Bioelectron. 2018, 102, 17-26, Trends Anal. Chem. 2022, 156, 116711).
Response: Thanks for the reviewer’s valuable suggestions. The novelty was emphasized together with the explanation, and all of the above mentioned recent reviews were cited. (The second and sixth references are the same in the above mentioned reviews, Talanta 2021, 224, 121794)
In the revised manuscript:
Though many reviews of MIPs about specific aspects have been reported, there are few reviews about the classification, preparation, and application [13-19]. As the rapid development and application of new MIPs, now is an appropriate time to summarize the recent progress [20-26].
Ref.
- Song, Z.H.; Li, J.H.; Lu, W.H.; Li, B.W.; Yang, G.Q.; Bi, Y.; Arabi, M.; Wang, X.Y.; Ma, J.P.; Chen, L.X. Molecularly imprinted polymers based materials and their applications in chromatographic and electrophoretic separations. Trends Anal. Chem., 2022, 146, 116504. DOI: 1016/j.trac.2021.116504.
- Moein, M.M. Advancements of chiral molecularly imprinted polymers in separation and sensor fields: A review of the last decade. Talanta, 2021, 224, 121794. DOI: 1016/j.talanta.2020.121794.
- Rutkowska, M.; Wasylka, J.P.; Morrison, C.; Wieczorek, P.P.; Namieśnik, J.; Marć, M. Application of molecularly imprinted polymers in an analytical chiral separation and analysis. Trends Anal. Chem., 2018, 102, 91-102. DOI: 1016/j.trac.2018.01.011.
- Ramanavicius, S.; Bubniene, U.S.; Ratautaite, V.; Bechelany, M.; Ramanavicius, A. Electrochemical molecularly imprinted polymer based sensors for pharmaceutical and biomedical applications. Pharm. Biomed. Anal., 2022, 215, 114739. DOI: 10.1016/j.jpba.2022.114739f.
- Wang, J.H.; Liang, R.M.; Qin, W. Molecularly imprinted polymer-based potentiometric sensors. Trends Anal. Chem., 2020, 130, 115980. DOI: 1016/j.trac.2020.115980.
- Ansari, S.; Masounm, S. Recent advances and future trends on molecularly imprinted polymer-based fluorescence sensors with luminescent carbon dots. Talanta, 2021, 223, 121411. DOI: 1016/j.talanta.2020.121411.
- Ahmad, O.S.; Bedwell, T.S.; Esen, C.; Cruz, A.G.; Piletsky, S.A. Molecularly imprinted polymers in electrochemical and optical sensors. Trends Biotechnol., 2019, 37, 294-309. DOI: 1016/j.tibtech.2018.08.009.
- Jancuzura, M.; Luliński, P.; Sobiech, M. Imprinting technology for effective sorbent fabrication: Current state-of-art and future prospects. Materials, 2021, 14, 1850. DOI: 3390/ma14081850.
- Parisi, O.I.; Francomano, F.; Dattilo, M.; Patitucci, F.; Prete, S.; Amone, F.; Puoci, F. The evolution of molecular recognition: From antibodies to molecularly imprinted polymers (MIPs) as artificial counterpart. Funct. Biomater., 2022, 13, 12. DOI: 10.3390/jfb13010012.
- Reville, E.K.; Sylvester, E.H.; Benware, S.J.; Negi, S.S.; Berda, E.B. Customizable molecular recognition: advancements in design, synthesis, and application of molecularly imprinted polymers. Chem., 2022, 13, 3387-3411. DOI: : 10.1039/d1py01472b.
- Cui, B.C.; Liu, P.; Liu, X.J.; Liu, S.Z.; Zhang, Z.H. Molecularly imprinted polymers for electrochemical detection and analysis: progress and perspectives. Mater. Res. Technol., 2020, 9, 12568-12584. DOI: 10.1016/j.jmrt.2020.08.052.
- Zhou, T.Y.; Ding, L.; Che, G.B.; Jiang, W.; Sang, L. Recent advances and trends of molecularly imprinted polymers for specifific recognition in aqueous matrix: Preparation and application in sample pretreatment. Trends Anal. Chem., 2019, 114, 11-28. DOI: 1016/j.trac.2019.02.028.
- Dabrowski, M.; Lach, P.; Cieplak, M.; Kutner, W. Nanostructured molecularly imprinted polymers for protein chemosensing. Bioelectron., 2018, 102, 17-26. DOI: 10.1016/j.bios.2017.10.045.
- Budnicka, M.; Sobiech, M.; Kolmsa, J.; Luliński, P. Frontiers in ion imprinting of alkali- and alkaline-earth metal ions e Recent advancements and application to environmental, food and biomedical analysis. Trends Anal. Chem., 2022, 156, 116711. DOI: 1016/j.trac.2022.116711.
- An important aspect to be considered is the fact that tables have not been included. Tables summarizing and highlighting the most important aspects of the articles found in the literature provide valuable and clear information to the Reader. This is extremely necessary in order to provide and demonstrate a suitable and complete revision of the literature. Please, add at least one item to the manuscript, for instance summarizing the application possibilities of imprinted polymers.
Response: Thanks for the reviewer’s valuable suggestions, seven tables were added according to the author’s comment.
- In my opinion, the Section 5 devoted to imprinted polymers for biomacromolecules should be extended since it focuses on the very important aspect of application, viz. therapy and diagnostics. Please, extend the discussion and complete the references (see: Acta Biomater. 2020, 101, 444-458, Trends Biotechnol. 2020, 38, 368-387, Theranostics 2022, 12, 2406-2426, Polymers 2019, 11, 2085, Trends Anal. Chem. 2021, 143, 116414, Biomedicines 2021, 9, 1923, Appl. Mater. Interfaces 2017, 9, 3006-3015, ACS Nano 2021, 15, 18214-18225, Appl. Mater. Interfaces 2016, 8, 5747-5751, Appl. Mater. Interfaces 2019, 11, 32431-32440).
Response: Thanks for the reviewer’s valuable suggestions. In section 5, the imprinted polymers for biomacromolecules were extended, and all of the above mentioned references were cited according to the author’s comment.
In the revised manuscript:
With the development of surface imprinting technology, especially the emergence of epitope imprinting technology, imprinted polymer have been widely used in the recognition and separation of biological macromolecules (Table 7) [10-12]. Instead of imprinting the whole biological macromolecules, imprinting exposed peptides is gaining popularity for its low cost and high stability [129-133].
To solve the problem of lack effective targeting for fluorescent conjugated polymer (FCP) in biological imaging, sialic acid (SA) was used as a template in the construction of FCP based MIPs [141]. The obtained SA MIPs showed enhance fluorescence intensity than that of NIPs, and exhibited selective staining for cancer cells. To modulate the adsorption and release performance of carrier, thermoresponsive EIPs was synthesized with thermal polymerization followed by chemical cross-linking [142]. The obtained EIPs could adsorb 46.6 mg/g of template protein, with a imprinting factor of 4.0. In addition, the template could capture the template at 45 °C, and release it at 4 °C. To achieve both precise targeting and drug delivery, a dual-template EIPs was fabricated for targets diagnosis and drug delivery of pancreatic cancer cells [143]. Modified epitope peptide (Glu-FH) and bleomycin (BLM) were used for the fabrication of dual-template EIPs, and the obtained EIPs not only showed obvious targeting effect, but also showed enhanced inhibiting to cancer cells. A sialic acid imprinted biodegradable nanoparticles based protein delivery was developed for targeted caner therapy [144]. With the loading of cytotoxic ribonuclease A (RNase A), the obtained EIPs showed specific tumor-targeting ability and high therapeutic efficacy.
Figure 16. Synthesis of PD-L1 peptide-imprinted composite [145]. PD-L1 peptide imprinted polymers was prepared by precipitation, with incorporation of merocyanine 540 (MC540)-grafted magnetic nanoparticles and green-emitting upconversion nanoparticles. The obtained composites could kill tumor cells precisely, with a enhancing efficacy of photodynamic therapy.
Ref.
129 Ali, M.M.; Zhu, S.J.; Amin, F.R.; Hussain, D.; Du, Z.X.; Hu, L.H. Molecular imprinting of glycoproteins: From preparation to cancer theranostics. Theranostics, 2022, 12, 2406-2426. DOI: 10.7150/thno.69189.
130 Bodoki, A.E.; Iacob, B.C.; Bodoki, E. Perspectives of molecularly imprinted polymer-based drug delivery systems in cancer therapy. Polymers, 2019, 11, 2085. DOI: 10.3390/polym11122085.
131 Dietl, S.; Sobek, H.; Mizaikoff, B. Epitope-imprinted polymers for biomacromolecules: Recent strategies, future challenges and selected applications. Trends Anal. Chem., 2021, 143, 116414. DOI: 10.1016/j.trac.2021.116414.
132 Piletsky, S.; Canfarotta, F.; Poma, A.; Bossi, A.M.; Piletsky, S. Molecularly imprinted polymers for cell recognition. Trends Biotechnol., 2020, 38, 368-387. DOI: 10.1016/j.tibtech.2019.10.002.
133 Vaneckova, T.; Bezdekova, J.; Han, G.; Adam, V.; Vaculovicova, M. Application of molecularly imprinted polymers as artifificial receptors for imaging. Acta Biomater., 2020, 101, 444-458. DOI: 10.1016/j.actbio.2019.11.007.
141 Liu, R.H.; Cui, Q.L.; Wang, C.; Wang, X.Y.; Yang, Y.; Li, L.D.; Preparation of sialic acid-imprinted fluorescent conjugated nanoparticles and their application for targeted cancer cell imaging. Appl. Mater. Interfaces, 2017, 9, 3006-3015. DOI: 10.1021/acsami.6b14320.
142 Li, S.W.; Yang, K.P.; Deng, N.; Min, Y.; Liu, L.K.; Zhang, L.H.; Zhang, Y.K. Thermoresponsive epitope surface-imprinted nanoparticles for specific capture and release of target protein from human plasma. Appl. Mater. Interfaces, 2016, 8, 5747-5751. DOI: 10.1021/acsami.5b11415.
143 Jia, C.; Zhang, M.; Zhang, Y.; Ma, Z.B.; Xiao, N.N.; He, X.W.; Li, W.Y.; Zhang, Y.K. Preparation of dual-template epitope imprinted polymers for targeted fluorescence imaging and targeted drug delivery to pancreatic cancer BxPC-3 cell. Appl. Mater. Interfaces, 2019, 11, 32431-32440. DOI: 10.1021/acsami.9b11533.
144 Lu, H.F.; Xu, S.X.; Guo, Z.C.; Zhao, M.H.; Liu, Z. Redox-responsive molecularly imprinted nanoparticles for targeted intracellular delivery of protein toward cancer therapy. ACS Nano, 2021, 15, 18214-18225. DOI: 10.1021/acsnano.1c07166.
145 Lin, C.C.; Lin, H.Y.; Thomas, J.L.; Yu, J.X.; Lin, C.Y.; Chang, Y.H.; Lee, M.H.; Wang, T.L. Embedded upconversion nanoparticles in magnetic molecularly imprinted polymers for photodynamic therapy of hepatocellular carcinoma. Biomedicines, 2021, 9, 1923. DOI: 10.3390/biomedicines9121923.
- In my opinion, selected older references (e.g. Refs. 4 and 12) as well as references in Chinese (e.g. Refs. 4 and 16) could be deleted and/or replaced by new ones, preferably provided in English. Unless it is highly necessary, Authors shall re-think reference to Master Thesis (Refs 27 and 28).
Response: Thanks for the reviewer’s valuable suggestions. In the revised manuscript, the old Ref. 4 was replaced by an recent international reference, but Ref.12 was reserved as it is very important and necessary. Chinese Refs. 4, 6, 16, 18 (corresponding to Refs. 4, 6, 30, 32 in the revised manuscript) were replaced by international references, but other Chinese Refs. 32, 52-55 (Refs. 47, 67-70 in the revised manuscript) has been reserved as there are few reports about the synthesis of MIPs with Electron Beam Radiation polymerization in English. In addition, all of the three MSc thesis 22, 27, 28 (Refs. 36, 41, 42 in the revised manuscript) have also been replaced with references from international journals.
Ref.
4 Ali, G.K.; Omer, K.M. Molecular imprinted polymer combined with aptamer (MIP-aptamer) as a hybrid dual recognition element for bio(chemical) sensing applications. Review. Talanta, 2022, 236, 122878. DOI: 10.1016/j.talanta.2021.122878.
6 Hasanah, A.N.; Safitri, N.; Zulfa, A.; Neli, N.; Rahayu, D. Factors affecting preparation of molecularly imprinted polymer and methods on finding template-monomer interaction as the key of selective properties of the materials. Molecules, 2021, 26, 5612. DOI: 10.3390/molecules26185612.
30 Akgönüllü, S.; Kiliç, S.; Esen, C.; Denizli, A. Molecularly imprinted polymer-based sensors for protein detection. Polymers, 2023, 15, 629. DOI: 10.3390/polym15030629.
32 Ayivi, R.D.; Adesanmi, B.O.; McLamore, E.S.; Wei, J.J.; Obare, S.O. Molecularly imprinted plasmonic sensors as nano-transducers: An effective approach for environmental monitoring applications. Chemosensors, 2023, 11, 203. DOI: 10.3390/chemosensors11030203.
36 Zhang, Y.; Zhao, G.L.; Han, K.Y.; Sun, D.N.; Zhou, N.; Song, Z.H.; Liu, H.T.; Li, J.H.; Li, G.S. Applications of molecular imprinting technology in the study of traditional chinese medicine. Molecules, 2023, 28, 301. DOI: 10.3390/molecules28010301.
41 Afzal, A.; Mujahid, A.; Schirhagl, R.; Bajwa, S.Z.; Latif, U.; Feroz, S. Gravimetric viral diagnostics: QCM based biosensors for early detection of viruses. Chemosensors, 2017, 5, 7. DOI: 10.3390/chemosensors5010007.
42 Ertürk, G.; Mattiasson, B. Molecular imprinting techniques used for the preparation of biosensors. Sensors, 2017, 17, 288. DOI: 10.3390/s17020288.
- Minor editorial issues: Ref. 51 -should be: Gadzala-Kopciuch, R., Ref. 40-should be Anal. Lett., Ref. 88-should be J. Chromatogr. Sci. and similarly Ref. 109.
Response: Thanks for the reviewer’s valuable suggestions, “Kopciuch, R.G.”, “Anal. Let.”, “J. Chromatogra. Sci.” were revised to “Gadzala-Kopciuch, R.”, “Anal. Lett.”, “J. Chromatogr. Sci.”

Round 2
Reviewer 1 Report
Authors have responded to all issues raised and adjusted the manuscript accordingly. Thus, I recommend the publication of the current version of this manuscript.
Reviewer 2 Report
The authors have improved their manuscript according to my comments.
The quality of english language was significantly improved.
Reviewer 3 Report
Review: polymers-2363975-R1.
Title: Preparation and application progress of imprinted polymers.
In this revised manuscript, the Authors have made corrections according to referee comments. In my opinion, the manuscript in current form could be considered for acceptance.